# LAGUNA: LAnguage Guided UNsupervised Adaptation with structured spaces

## Abstract

Unsupervised domain adaptation remains a critical challenge in enabling knowledge transfer of models across domains. Existing methods struggle to balance the need for domain-invariant representations with preserving domain-specific features. This is often caused by alignment approaches that impose the projection of different domain samples close to each other in latent spaces, despite drastic differences. We introduce LAGUNA - LAnguage Guided UNsupervised Adaptation, a novel approach that shifts the focus from aligning representations in absolute coordinates to aligning the relative positioning of equivalent concepts in latent spaces. LAGUNA defines a domain-agnostic structure on the geometric relationships between class labels in the language space and guides the adaptation, ensuring that the organization of samples in visual space reflects reference inter-class relationships while preserving domain-specific characteristics. Remarkably, LAGUNA surpasses previous work in 18 different adaptation scenarios across four diverse image and video datasets with average accuracy improvements of up to $+3.32\%$ on DomainNet, $+5.75\%$ on GeoPlaces, $+4.77\%$ on GeoImnet, and $+1.94\%$ mean class accuracy improvement on EgoExo4D.

## 1 Introduction

Domain shift challenges trained models to generalize across scenarios with differing distributions and presents a significant hurdle for supervised learning in computer vision (Luo et al., 2018). While fine-tuning with labeled data from the target domain can mitigate this problem, obtaining such labels often proves prohibitive. Unsupervised domain adaptation (UDA) (Bousmalis et al., 2016) offers a compelling alternative, enabling knowledge transfer to novel domains without relying on expensive labeled data, which has gained significant attention (Wilson and Cook, 2020), promising cost-effective solutions for real-world applications prone to domain shift. The typical UDA setting considers the availability of a labeled source domain and an unlabeled target domain. In general, source and target domains are semantically equivalent but drawn from distinct data distributions, e.g., real images versus clipart of chairs, TVs, and mugs. Thus, the main challenge of UDA is to effectively mitigate the distribution shift between domains, which is often addressed by reducing the distribution discrepancy of source and target representation spaces either by minimizing some discrepancy measure (Saito et al., 2018b; Tan et al., 2020), using adversarial learning (Wei et al., 2021; Li et al., 2020), aligning data around centroids (Li et al., 2021), or leveraging pseudo-labels (Kalluri et al., 2024; Li et al., 2021). These methods aim to align source and target representation spaces in a shared coordinate system, pushing feature vectors of equivalent semantic concepts close to each other in the embedding space, which may happen at the expense of the representation power of individual domain-specific spaces. For instance, bright colors and rounded shapes might be important to encode for representing a clipart, while nuanced shadows, reflections, and textures might be important to represent a real image. As a result, the aligned space may become overly generic, correctly encoding only a subset of the data (see Figure 1(left)). Recent work (Moschella et al., 2023) showed that semantically equivariant representation spaces of similarly trained neural networks exhibit distinct representation spaces with matching geometrical structures. For instance, two data points $(x_1, x_2)$ may be mapped to distinct vector pairs in the two representation spaces $(v_1^1, v_2^1)$ and $(v_1^2, v_2^2)$ (e.g., $||v_1^1 - v_1^2||_2 >> 0$ and $||v_2^1 - v_2^2||_2 >> 0$), while angles between the two

Figure 1: **Left**: Existing UDA approaches align source and target spaces in absolute coordinates, potentially overlooking domain-specific characteristics and resulting in partial alignment. **Right**: LAGUNA aligns spaces in relative terms, preserving distinct absolute coordinates (e.g., circles in source and target) while matching angles $\theta_i^s$, $\theta_i^t$ between data points to a reference structure $\theta_i^l$ ($\theta_i^s \sim \theta_i^l \sim \theta_i^t$), encouraging similar geometric-semantic relations.

pairs will be similar in their own representation spaces (e.g., $\angle(v_1^1, v_2^1) \sim \angle(v_1^2, v_2^2)$) (Moschella et al., 2023). This suggests that pushing representation spaces to overlap in absolute coordinates, as done in current domain adaptation approaches, is not necessary to obtain equivariant representation spaces.

Based on this observation, we introduce LAGUNA - LAnguage Guided UNsupervised Adaptation, a novel approach to UDA which guides source and target spaces to develop semantic-geometric inter-relationships reflecting the structure of a reference space (Fig. 1 (right)). As shown in recent works, language can provide a semantic space agnostic to the nuances of visual observations, enabling robust zero-shot generalization (Radford et al., 2021) and supporting domain robustness (Dunlap et al., 2022; Wang et al., 2024), hence we choose language to build our reference structured space in LAGUNA. This assumes the availability of both source and target samples of text descriptions, which can be generated from captioning models (Li et al., 2023) or collected from the web at a fraction of the cost of human labeling (Kalluri et al., 2024). LAGUNA employs a 3-stage approach to structurally align source and target representation spaces while allowing domains to preserve typical patterns (Figure 2). In Stage 1, textual class labels are mapped to a domain-agnostic reference space representing their semantic relationships. In Stage 2, a language model is trained to map captions to the reference latent space, providing pseudo-labels for target samples. In Stage 3, a cross-domain classifier is trained to encourage domain-specific representations to follow the reference structure.

Experiments demonstrate LAGUNA's superiority over existing SOTA methods across 18 different domain adaptation scenarios sourced from four diverse image and video datasets, with gains of up to $+3.32\%$ on DomanNet (Peng et al., 2018), $+5.75\%$ on GeoPlaces (Kalluri et al., 2023), $+4.77\%$ on GeoImnet (Kalluri et al., 2023) and $+1.94\%$ mean per class accuracy on EgoExo4D (Grauman et al., 2023). We further report ablations to analyze the specific contributions of our design choices and compare LAGUNA's performance against SOTA zero-shot massive MLLMs.

In sum, our main contributions are: 1) we investigate using relative representations for UDA, showing its advantages w.r.t absolute alignment; 2) we propose LAGUNA, a method which learns a cross-domain classifier where source and target spaces are distinct yet aligned to a reference structure; 3) through extensive ablations and comparisons with SOTA, we show the superiority of LAGUNA.

## 2 RELATED WORK

**UDA in Computer Vision.** UDA seeks to transfer knowledge from a labeled source domain to an unlabeled target domain (Ben-David et al., 2006; Ganin and Lempitsky, 2015; Long et al., 2018). This is tackled through various approaches, prominently, discrepancy-based methods that minimize distribution differences using techniques like MMD (Tan et al., 2020; Long et al., 2017b; Sun and Saenko, 2016; Li et al., 2020), and adversarial learning (Bousmalis et al., 2016; Saito et al., 2018a; Chen et al., 2019a; Wei et al., 2021; Long et al., 2018; 2017a). Other approaches also investigated class-conditional distributions alignment (Luo et al., 2019; Xie et al., 2018; Wei et al., 2021), clustering similar instances across domains (Deng et al., 2019; Liu et al., 2021a; Kalluri and Chandraker, 2022), instance-specific adaptation (Sharma et al., 2021; Kalluri et al., 2022; Wang et al., 2022), and self-training leveraging pseudo-labeling to improve target domain performance (French et al., 2018; Liu et al., 2021a; Sun et al., 2022; Kalluri et al., 2024). More recently, transformer-based approaches utilize patch composition for domain-invariant representations through intermediate domains (Zhu

et al., 2023). Video domain adaptation tackles the unique challenges of temporal dynamics and consistency in videos (Chen et al., 2019b; Wei et al., 2023; Sahoo et al., 2021) with a particular focus on adapting models across the exocentric and egocentric domains (Quattrocchi et al., 2023; Huang et al., 2024; Xue and Grauman, 2023). Existing approaches sought to align source and target representation spaces in absolute coordinates, often falling short in bridging complicated forms of domain shift (Kalluri et al., 2023; Prabhu et al., 2022). Contrarily, LAGUNA aligns representations in relative terms, encouraging source and target visual spaces to share a similar structure as a reference space derived from language while allowing them to encode domain-specific peculiarities.

**Language Guidance in Vision UDA.** Vision-language models like CLIP (Radford et al., 2021) have shown promising zero-shot transfer capabilities (Devillers et al., 2021). However, when fine-tuned to a specific scenario (Pham et al., 2021; Andreassen et al., 2021), they lose the ability to generalize to new domains (Kumar et al., 2022; Wortsman et al., 2021). To address this, recent works leveraged textual information to bridge the gap between domains (Dunlap et al., 2022; Wang et al., 2024; Huang et al., 2023; Min et al., 2022; Gokhale et al., 2020; Goyal et al., 2022). In particular, the seminal work of (Kalluri et al., 2024) proposed to use textual captions to provide pseudo-labels for the target domain and addressed UDA by training a joint classifier on source and target domains. Similarly to Kalluri et al. (2024), we generate pseudo-labels from textual captions, but, rather than seeking to align the visual representation spaces of source and target domains to language in an absolute reference frame, LAGUNA uses language to guide a *relative* alignment between source and target visual spaces.

**Relative Encodings.** Recent work (Moschella et al., 2023) has shown that equivalent latent spaces of similarly trained networks tend to be misaligned in absolute terms but share similar internal geometric relationships. As a result, using relative encodings, obtained as similarity values of data points w.r.t a predefined set of anchors, allows to perform zero-shot model stitching (Moschella et al., 2023), translate representation spaces across models (Maiorca et al., 2023) or modalities (Norelli et al., 2023). As shown in (Cannistraci et al., 2024), predefined invariances can be incorporated into the learned representation to enable specific forms of relative representations. In (Diko et al., 2024) relative encodings were used to tackle the downstream task of action anticipation. LAGUNA builds on relative encodings to represent the semantic inter-relations between classes in a domain-agnostic language-based reference space and support the development of aligned yet specialized source and target domains.

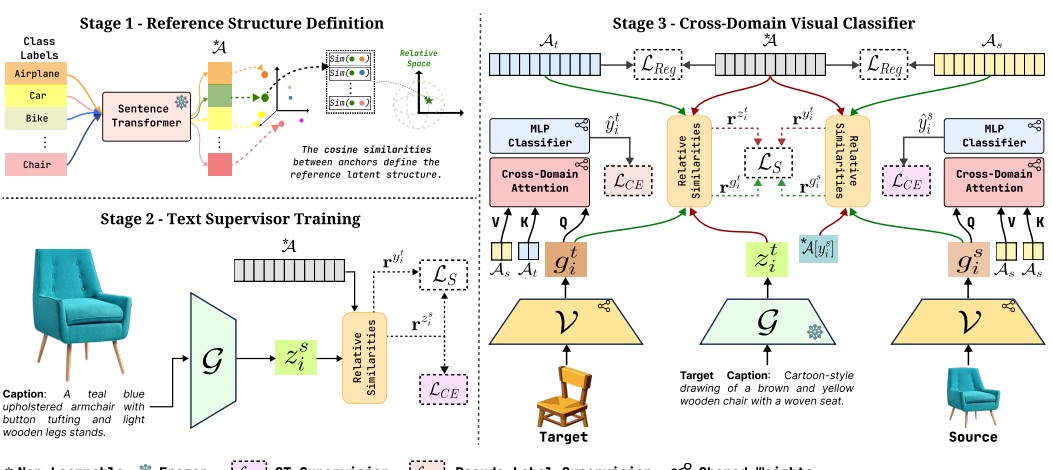

Figure 2: The 3-stage architecture. (1) We define domain-agnostic semantic anchors $\mathcal{A}$. (2) A language model $\mathcal{G}$ generates pseudo-labels for target data, trained with structural loss $\mathcal{L}_S$ and cross-entropy loss $\mathcal{L}_{CE}$. (3) A visual classifier is trained using an encoder $\mathcal{V}$ to extract features $g_i^s$ and $g_i^t$. We align visual-anchor similarities with text-anchor similarities using $\mathcal{L}_S$, learnable anchors ($\mathcal{A}_s$, $\mathcal{A}_t$), and textual representations ($z_i^t$, $^*A[y_i^S]$). A Cross-Domain Attention layer grounds visual features using $\mathcal{A}_s$, and an MLP classifier is trained with $\mathcal{L}_{CE}$ and regularized by $\mathcal{L}_{Reg}$.

## 3 METHOD

LAGUNA works in 3 stages, as shown in Fig. 2. First, we create a language-based reference structure using anchor points to guide representation learning. Second, we train a language model to map image/video captions to class categories, generating pseudo-labels for unlabeled target data while keeping text embeddings aligned to the reference structure. Finally, we train a cross-domain visual classifier that learns domain-specific action anchors structured similarly to our reference framework.

### 3.1 PROBLEM SETUP

We follow the formulation of Kalluri et al. (2024) and assume a labeled source domain $\mathcal{X}_s$ : $\{(x_i^s, y_i^s, l_i^s)\}_{i=1}^{N_s}$, where samples $x_i^s$ are paired with labels $y_i^s$ and captions $l_i^s$, whereas the target domain $\mathcal{X}_t : \{x_i^t, l_i^t\}_{i=1}^{N_t}$ contains unlabeled samples $x_i^t$ paired with language descriptions $l_i^t$. These textual descriptions can be readily obtained from associated metadata or generated using image-to-text models (Li et al., 2023). $N_s$ and $N_t$ represent the number of source and target samples, respectively, and the two domains share the same high-level semantics and number of classes $N_c$.

### 3.2 STAGE 1 - REFERENCE STRUCTURE DEFINITION

Assuming shared classes across domains, Stage 1 (Fig. 2) builds a language-based, domain-agnostic reference structure as a set of vectors $\mathcal{A} \in \mathbb{R}^{N_c \times D_l}$. We obtain each vector by encoding the $N_c$ class label names using a pre-trained *SentenceTransformer* model with output dimensionality $D_l$ trained for semantic similarity (Reimers and Gurevych, 2019). Following the relative representations literature (Moschella et al., 2023), we treat these vectors in $\mathcal{A}$ as reference *anchors*. For any vector $v \in \mathbb{R}^{D_l}$, we define its relative encoding with respect to anchors $\mathcal{A}$ as $\mathbf{r}^v = rel(v, \mathcal{A}) = [\cos(v, \mathcal{A}[1]), \dots, \cos(v, \mathcal{A}[N_c])]$, where $\cos(\cdot, \cdot)$ represents cosine similarity and $\mathcal{A}[i]$ is the anchor for class $i$. This vector $\mathbf{r}^v$ captures the geometric and semantic relationships between $v$ and our reference language anchors $\mathcal{A}$. We define the complete set of reference affinities as $\mathbf{r}^{\mathcal{A}} = [rel(\mathcal{A}[1], \mathcal{A}), \dots, rel(\mathcal{A}[N_c], \mathcal{A})]$, where each class anchor $\mathcal{A}[i]$ is represented by its relationships to all other anchors in $\mathcal{A}$. During Stage 3's cross-domain visual classifier training, these encodings enforce the learned latent space to follow the same structure induced by $\mathcal{A}$.

### 3.3 STAGE 2: TRAINING OF THE LANGUAGE SUPERVISOR

Similar to (Kalluri et al., 2024), we train a language model ($\mathcal{G}$) from captions to provide pseudo-labels for target samples, which have no class labels. In addition, we also use $\mathcal{G}$ to learn textual representations semantically structured as the domain-agnostic anchors $\mathcal{A}$, useful to encourage alignment to the reference structure. Hence, rather than training a regular classifier, we directly supervise $\mathcal{G}$ to provide representations that are 1) geometrically aligned to $\mathcal{A}$ and 2) suitable for predicting class labels. Specifically, given a pair of source caption-label $(l_i^s, y_i^s)$, $\mathcal{G}$ processes $l_i^s$ to produce a vector representation $z_i^s = \mathcal{G}(l_i^s)$. Next, we encourage the vector representation $z_i^s$ to be geometrically aligned to $\mathcal{A}[y_i^s]$ corresponding to ground truth action $y_i^s$. To do so, we compute $\mathbf{r}^{z_i^s} = rel(z_i^s, \mathcal{A})$, the relative encoding of $z_i^s$, and supervise it to be similar to $\mathbf{r}^{y_i^s} = rel(\mathcal{A}[y_i^s], \mathcal{A})$, the relative encoding of the anchor $\mathcal{A}[y_i^s]$, with the following structure-preserving loss:

$$\mathcal{L}_S = |\mathbf{r}^{z_i^s} - \mathbf{r}^{y_i^s}|. \tag{1}$$

This loss encourages the encodings $z_i^s$ of text descriptions $l_i^s$ associated with $y_i^s$ to preserve the same geometrical associations as $\mathcal{A}[y_i^s]$, encouraging the latent space learned by $\mathcal{G}$ to mirror the structure defined by $\mathcal{A}$. To further favor alignment to $\mathcal{A}$, rather than employing a classification head, we predict class probabilities directly by Softmax-normalizing relative encodings:

$$p_j^{z_i^s} = \frac{e^{\mathbf{r}_j^{z_i^s}}}{\sum_k e^{\mathbf{r}_k^{z_i^s}}}. \tag{2}$$

Finally, we train the model using a combined loss $\mathcal{L}$:

$$\mathcal{L} = \lambda_1 \mathcal{L}_{CE}(p^{z_i^s}, y_i^s) + \lambda_2 \mathcal{L}_S(\mathbf{r}^{z_i^s}, \mathbf{r}^{y_i^s}), \tag{3}$$

where $\mathcal{L}_{CE}$ is the cross-entropy loss, while $\lambda_1$ and $\lambda_2$ are hyperparameter weights to calibrate the magnitude of each loss. We refer to the label predicted by $\mathcal{G}$ from $l_i^t$ as $\overline{y}_i^t$.

### 3.4 STAGE 3: CROSS-DOMAIN VISUAL CLASSIFIER

Stage 3 trains a cross-domain visual classifier aligning representations extracted through a visual encoder $\mathcal{V}$ to the structure imposed by $\mathcal{A}$. We allow each domain to develop its own latent space, but we encourage both spaces to be aligned to two sets of learnable anchors $\mathcal{A}_t \in \mathbb{R}^{N_c \times D_v}$ and $\mathcal{A}_s \in \mathbb{R}^{N_c \times D_v}$ specific to the target and source domain respectively. We further ensure that the structures of $\mathcal{A}_t$ and $\mathcal{A}_s$ are aligned to that of $\mathcal{A}$ through supervision provided by $y_i^s$ in the source domain and the text supervision provided by $\mathcal{G}$ in the target domain in the form of $z_i^t$ and $\overline{y}_i^t$. The source and target datasets are merged ($\hat{\mathcal{X}} = \mathcal{X}_t + \mathcal{X}_s$) and used for training with a total of $M = N_s + N_t$ samples. Source samples comprise $(x_i^s, l_i^s, y_i^s)$ triplet, whereas target samples $(x_i^t, l_i^t, \overline{y}_i^t)$ use $\overline{y}_i^t$ in absence of $y_i^t$. Target and source images are processed by $\mathcal{V}$ to obtain latent representations $g_i^t$ and $g_i^s$, while target captions $l_i^t$ are processed by $\mathcal{G}$ to obtain latent representations $z_i^t$ as follows:

$$g_i^t = \mathcal{V}(x_i^t), \qquad z_i^t = \mathcal{G}(l_i^t), \qquad g_i^s = \mathcal{V}(x_i^s). \tag{4}$$

#### 3.4.1 STRUCTURE LEARNING

To ensure the learned visual representations adhere to the structure defined by $\mathcal{A}$, LAGUNA employs a supervised learning approach based on relative encodings. For target samples, we first compute $\mathbf{r}^{z_i^t} = rel(z_i^t, \mathcal{A})$. For source samples, we compute the relative encoding of the anchor associated with the ground truth class $\mathcal{A}[y_i^s]$: $\mathbf{r}^{y_i^s} = rel(\mathcal{A}[y_i^s], \mathcal{A})$. These encodings represent the relative positions of textual representations with respect to the reference space $\mathcal{A}$ and are not trainable as $\mathcal{G}$ is fixed. On the visual side, we compute the relative encodings of $g_i^t$ and $g_i^s$ with respect to the learnable anchors $\mathcal{A}_t$ and $\mathcal{A}_s$, i.e., $\mathbf{r}^{g_i^t} = rel(g_i^t, \mathcal{A}_t)$ and $\mathbf{r}^{g_i^s} = rel(g_i^s, \mathcal{A}_s)$. We encourage the visual relative encodings ($\mathbf{r}^{g_i^t}, \mathbf{r}^{g_i^s}$) to match the text relative encodings ($\mathbf{r}^{z_i^t}, \mathbf{r}^{y_i^s}$) with an L1 loss:

$$\mathcal{L}_S = \begin{cases} |\mathbf{r}^{g_i^t} - \mathbf{r}^{z_i^t}|, & \textit{if Target Domain} \\ |\mathbf{r}^{g_i^s} - \mathbf{r}^{y_i^s}|, & \textit{otherwise} \end{cases}. \tag{5}$$

This loss encourages $g_i^t$ and $g_i^s$ to maintain relationships with their respective domain-specific visual anchors that mirror those expressed by $\mathcal{A}$. This process supervises both the learnable anchors $\mathcal{A}_s, \mathcal{A}_t$ and the visual encoder $\mathcal{V}$. However, since this loss alone might lead to the collapse of anchor representations into a small region, hindering classification decision boundaries, we introduce a volume (spread) regularization loss. This loss treats each set of anchors as a multidimensional parallelotope whose volume in the latent space is measured by the determinant of its Gram matrix. Given the Gram matrices for the three sets of anchors:

$$\gamma = \mathcal{A}\mathcal{A}^T, \qquad \gamma_s = \mathcal{A}_s\mathcal{A}_s^T, \qquad \gamma_t = \mathcal{A}_t\mathcal{A}_t^T, \tag{6}$$

we encourage the volume of each domain-specific parallelotope to be approximately equal to that of $\mathcal{A}$ with the following loss:

$$\mathcal{L}_{Reg} = |logDet(\gamma_t) - logDet(\gamma)| + |logDet(\gamma_s) - logDet(\gamma)|. \tag{7}$$

We use log-determinants for numerical stability. This regularization loss ensures that visual anchors occupy a volume similar to that of $\mathcal{A}$, preventing collapse in the representation space.

#### 3.4.2 CLASSIFICATION TRAINING

We perform classification training concurrently with structure learning. To ensure that source and target visual representations are grounded in the structure imposed by $\mathcal{A}$, while still being free to capture domain-specific nuances, we propose a novel cross-domain attention layer that aims to ground $g_i^t$ and $g_i^s$ to the structure of source anchors $\mathcal{A}_s$. We include two Cross-Domain attention layers sharing the same weights for the target and source branches of LAGUNA (See Fig. 2). In the target branch, we use $g_i^t$ as queries, target anchors $\mathcal{A}_t$ as keys, and source anchors $\mathcal{A}_s$ as values. The output is summed to $g_i^t$ with a residual connection to include both domain-specific and cross-domain information in the final representation:

$$f_i^t = \text{Attention}(Q = g_i^t, K = \mathcal{A}_t, V = \mathcal{A}_s) + g_i^t \tag{8}$$

A similar processing is applied to the source encoding $g_i^s$:

$$f_i^s = \text{Attention}(Q = g_i^s, K = \mathcal{A}_s, V = \mathcal{A}_s) + g_i^s \tag{9}$$

Crucially, the cross-domain attention layer constructs an attention map by leveraging the domain-specific relations between the encoded representations and the anchors (i.e., $QK^T$). These relations are expected to be similar across domains due to the influence of the structural loss $\mathcal{L}_S$. This process results in a unified representation since the output of the attention layer always depends on values coming from the source anchors $\mathcal{A}_s$. The residual connection included in the attention layer is introduced to account for domain-dependent class characteristics, which results in the feature vectors $f_i^s$ and $f_i^t$. Finally, an MLP classification head processes $f_i^t$ and $f_i^s$ to output predictions $\hat{y}_i^t$ and $\hat{y}_i^s$. The MLP has shared weights across domains and is optimized with a cross-entropy loss using $y_i^s$ for source examples and $\overline{y}_i^t$ in the case of target samples. The overall training objective is:

$$\mathcal{L} = \lambda_1 \mathcal{L}_{CE}(\hat{y}_i, y_i^*) + \lambda_2 \mathcal{L}_S(\mathbf{r}^*, \mathbf{r}^{g_i}) + \lambda_3 \mathcal{L}_{Reg}(\gamma, \gamma_*), \tag{10}$$

where $\lambda_1$, $\lambda_2$, and $\lambda_3$ are hyperparameter weights to calibrate the magnitude of each loss value, and '*' signifies that the argument is domain-dependent[1].

## 4 EXPERIMENTS

### 4.1 EXPERIMENTAL SETUP

**Datasets.** We evaluate LAGUNA on four comprehensive UDA benchmarks: DomainNet (Peng et al., 2018), GeoPlaces (Kalluri et al., 2023) GeoImnet (Kalluri et al., 2023), and Ego2Exo (Kalluri et al., 2024). DomainNet consists of $400K$ images across $345$ classes and is used to evaluate performance in 12 adaptation settings. GeoImnet and GeoPlaces, subsets of the GeoNet dataset, contain over 750K images, and focus on geographic disparities for objects (GeoImNet with $600$ classes) and scenes (GeoPlaces with $205$ classes). Finally, Ego2Exo is a subset of EgoExo-4D (Grauman et al., 2023) curated for domain adaptation. It involves a total of 9086 videos and 24 categories. DomainNet and GeoNet include per-image captions derived from BLIP-2 (Li et al., 2023) and web metadata, respectively, while Ego2Exo provides video descriptions from the original EgoExo-4D.
**Model and training details.** We use a pre-trained *SentenceTransformer* (Reimers and Gurevych, 2019) for Stage 1 and a BERT base model (Sanh et al., 2019) as text supervisor (Stage 2). The latter is trained on the source domain of each scenario for 5 epochs with a batch size of 64, a lr of 1e-4, and the AdamW (Loshchilov, 2017) optimizer. The visual encoders (Stage 3) are tailored to each dataset for fair comparisons with existing approaches. For DomainNet, we adapt the Swin-B and ViT-B backbones (Liu et al., 2021b; Dosovitskiy et al., 2020). For GeoImnet and GeoPLaces, we utilize the ViT-B backbone. For Ego2Exo, we employ pre-extracted Omnivore-L features (Grauman et al., 2023). All vision models undergo joint training with the other network components for 10 epochs with a batch size of 32, an initial learning rate of $1e-4$, and cosine scheduling. We set the loss weights to $\lambda_1$=1.0, $\lambda_2$=0.1, and $\lambda_3$=0.001 to normalize the magnitudes of loss components.

### 4.2 RESULTS

We benchmark our model against various language-free UDA methods that have reported results on the considered datasets (Zhu et al., 2023; Wei et al., 2021; Kalluri et al., 2022; Zhang et al., 2019; Chen et al., 2022; Xu et al., 2021). Additionally, we also compare with previous approaches using language guidance (Kalluri et al., 2024; Radford et al., 2021; Ge et al., 2025; Lai et al., 2023; Li et al., 2024). Following the evaluation protocols of each benchmark, we report accuracy for Domain-Net, GeoImNet, and GeoPlaces, and per-class mean accuracy for Ego2Exo.
**DomainNet.** Table 1 presents the results on DomainNet, encompassing 12 UDA scenarios across four distinct domains: Real (R), Clipart (C), Sketch (S), and Painting (P). LAGUNA is compared against previous SOTA methods utilizing the Swin-B and ViT-B backbones as originally adopted by the compared methods to ensure fair comparison. LAGUNA significantly outperforms previous methods in 12 scenarios, with an average performance increase of 3.32% for Swin-B. For ViT-B, LAGUNA leads in 11 of 12 scenarios, boosting performance by an average of 2.51%. Importantly, LAGUNA achieves SOTA ViT-B performance without relying on extensive CLIP pretraining and domain-aware prompting, unlike previous models (Lai et al., 2023; Li et al., 2024; Ge et al., 2025).
**GeoImnet & GeoPlaces.** Table 2 presents the results on GeoImnet and GeoPlaces covering adaptation between the USA (U) and Asia (A) geographical domains. LAGUNA consistently outperforms

---

[1]For example, for $\mathcal{L}_{CE}$ the ground truth can either be $y_i^s$ or $\overline{y}_i^t$.

| Source→ | Real | | | Clipart | | | Sketch | | | Painting | | | Avg. |
|---|---|---|---|---|---|---|---|---|---|---|---|---|---|
| Target→ | C | S | P | R | S | P | R | C | P | R | C | S | |
| *Language-free Approaches* | | | | | | | | | | | | | |
| Source Only | 63.02 | 49.47 | 60.48 | 70.52 | 56.09 | 52.53 | 70.42 | 65.91 | 54.47 | 73.34 | 60.09 | 48.25 | 60.38 |
| MDD (Zhang et al., 2019) | 52.80 | 41.20 | 47.80 | 52.50 | 42.10 | 40.70 | 54.20 | 54.30 | 43.10 | 51.20 | 43.70 | 41.70 | 47.11 |
| SCDA (Li et al., 2021) | 54.00 | 42.50 | 51.90 | 55.00 | 44.10 | 39.30 | 53.20 | 55.60 | 44.70 | 56.20 | 44.10 | 42.00 | 48.55 |
| SSRT-B (Sun et al., 2022) | 69.90 | 58.90 | 66.00 | 75.80 | 59.80 | 60.80 | 73.20 | 70.60 | 62.20 | 71.40 | 61.70 | 55.20 | 65.41 |
| MemSAC (Kalluri et al., 2022) | 63.49 | 42.14 | 60.32 | 72.33 | 54.92 | 46.14 | 73.46 | 68.04 | 52.75 | 74.42 | 57.79 | 43.57 | 59.11 |
| CDTrans (Xu et al., 2021) | 66.20 | 52.90 | 61.50 | 72.60 | 58.10 | 57.10 | 72.50 | 69.00 | 59.00 | 72.10 | 62.90 | 53.90 | 63.16 |
| PMTrans (Zhu et al., 2023) | 74.10 | 61.10 | 70.00 | 79.30 | 63.70 | 62.70 | 77.50 | 73.80 | 62.60 | 79.80 | 69.70 | 61.20 | 69.63 |
| *UDA with Language Guidance* | | | | | | | | | | | | | |
| TextMatch (Kalluri et al., 2024) | 71.36 | 64.30 | 65.32 | 81.25 | 65.65 | 64.85 | 81.09 | 72.65 | 63.94 | 81.08 | 70.84 | 64.17 | 70.64 |
| nGramMatch (Kalluri et al., 2024) | 68.92 | 59.82 | 63.15 | 76.35 | 61.72 | 62.87 | 76.35 | 69.28 | 62.51 | 76.04 | 68.52 | 60.52 | 67.17 |
| LaGTran (Kalluri et al., 2024) | 77.30 | 68.25 | 67.35 | 81.31 | 67.03 | 66.81 | 80.78 | 75.62 | 68.08 | 79.23 | 73.80 | 63.44 | 72.41 |
| LAGUNA | 80.34 | 70.68 | 71.92 | 83.07 | 69.51 | 70.59 | 83.34 | 79.71 | 70.51 | 83.32 | 77.47 | 68.32 | 75.73 |
| Improvement | +3.04 | +2.43 | +1.92 | +1.82 | +2.48 | +3.78 | +2.25 | +4.09 | +2.43 | +2.24 | +3.67 | +4.15 | +3.32 |
| CLIP* (Radford et al., 2021) | 72.39 | 60.90 | 66.81 | 81.37 | 60.90 | 66.81 | 81.37 | 72.39 | 66.81 | 81.37 | 72.39 | 69.90 | 70.38 |
| DAPrompt* (Ge et al., 2025) | 73.90 | 65.90 | 70.40 | 84.90 | 65.80 | 70.20 | 84.60 | 73.50 | 69.90 | 84.90 | 73.80 | 65.80 | 73.80 |
| PADCLIP (Lai et al., 2023) | 76.40 | 67.50 | 72.70 | 84.20 | 68.10 | 71.10 | 83.60 | 76.30 | 71.70 | 83.50 | 75.40 | 67.20 | 74.81 |
| LaGTran† (Kalluri et al., 2024) | 76.33 | 69.11 | 71.22 | 84.38 | 68.48 | 71.33 | 84.31 | 75.72 | 71.98 | 84.71 | 75.71 | 68.94 | 75.18 |
| UniMoS* (Li et al., 2024) | 77.50 | 68.00 | 72.50 | 86.00 | 68.50 | 72.30 | 85.90 | 77.80 | 72.60 | 85.80 | 77.20 | 68.20 | 76.03 |
| LAGUNA | 79.53 | 73.18 | 74.24 | 86.04 | 73.54 | 74.98 | 85.81 | 80.24 | 74.43 | 86.57 | 79.89 | 74.11 | 78.54 |
| Improvement | +2.03 | +4.07 | +1.74 | +0.04 | +5.03 | +2.68 | -0.09 | +2.44 | +1.83 | +0.77 | +2.69 | +5.91 | +2.51 |

(Swin-B for the first two blocks; ViT-B for the last two blocks.)

Table 1: DomainNet dataset results for domain adaptation. Includes 4 domains: Real, Clipart, Sketch, and Painting, for a total of 12 domain adaptation scenarios. The best results are reported in **bold** while the second best are underlined.

| Model | GeoImnet | | GeoPlaces | | Avg. |
|---|---|---|---|---|---|
| | U→A | A→U | U→A | A→U | |
| Source Only | 52.46 | 51.91 | 44.90 | 36.85 | 46.53 |
| CDAN (Long et al., 2018) | 54.48 | 53.87 | 42.88 | 36.21 | 46.86 |
| MemSAC (Kalluri et al., 2022) | 53.02 | 54.37 | 42.05 | 38.33 | 46.94 |
| ToAlign (Wei et al., 2021) | 55.67 | 55.92 | 42.32 | 38.40 | 48.08 |
| MDD (Zhang et al., 2019) | 51.57 | 50.73 | 42.54 | 39.23 | 46.02 |
| DALN (Chen et al., 2022) | 55.36 | 55.77 | 41.06 | 40.41 | 48.15 |
| PMTrans (Zhu et al., 2023) | 56.76 | 57.60 | 46.18 | 40.33 | 50.22 |
| *UDA with Language Guidance* | | | | | |
| TextMatch (Kalluri et al., 2024) | 49.68 | 54.82 | 53.06 | 50.11 | 51.92 |
| nGramMatch (Kalluri et al., 2024) | 49.53 | 51.02 | 51.70 | 49.87 | 50.93 |
| LaGTran (Kalluri et al., 2024) | 63.67 | 64.16 | 56.14 | 57.02 | 60.24 |
| LAGUNA | 67.39 | 69.97 | 61.15 | 63.51 | 65.40 |
| Improvement | +3.72 | +5.81 | +5.01 | +6.49 | +5.16 |

Table 2: Results on GeoImnet and GeoPlaces with 4 adaptation scenarios using ViT-B. Best results are in **bold**, whereas second-best are underlined.

| Model | Ego→Exo | Exo→Ego | Avg. |
|---|---|---|---|
| Source Only | 8.39 | 15.66 | 12.03 |
| TA3N (Chen et al., 2019b) | 6.92 | 27.95 | 17.44 |
| TransVAE (Wei et al., 2023) | 12.06 | 23.34 | 17.70 |
| *Zero-shot Video Recognition* | | | |
| EgoVLP (Lin et al., 2022) | 5.89 | 19.35 | 12.62 |
| LaVILA (Zhao et al., 2022) | 5.86 | 23.16 | 14.51 |
| *UDA with Language Guidance* | | | |
| TextMatch (Kalluri et al., 2024) | 10.36 | 13.57 | 11.97 |
| nGramMatch (Kalluri et al., 2024) | 11.50 | 15.46 | 13.98 |
| LaGTran (Kalluri et al., 2024) | 12.34 | 30.76 | 21.55 |
| LAGUNA | 13.52 | 33.45 | 23.49 |
| Target Supervised (oracle) | 18.08 | 35.12 | 26.60 |
| Improvement | +1.18 | +2.69 | +1.94 |

Table 3: Results on the Ego2Exo benchmark. Best results are in **bold**, second best are underlined. All methods use Omnivore-L features except EgoVLP and LaVILA.

previous methods, achieving improvements of +3.72% (U→A) and +5.81% (A→U) on GeoImnet, and +5.01% (U→A) and +6.49% (A→U) on GeoPlaces, for an average improvement of +5.26%. **Ego2Exo.** LAGUNA demonstrates consistent gains also on the challenging Ego2Exo benchmark, introduced in (Kalluri et al., 2024). LAGUNA surpasses prior work, achieving improvements of +1.18% (Ego→Exo) and +2.69% (Exo→Ego), for an overall average gain of +1.94% (note that this is per-class mean accuracy), narrowing the gap with the oracle. These consistent gains across 18 adaptation scenarios highlight the significance of LAGUNA w.r.t prior UDA approaches.

## 4.3 ABLATION STUDY

In this section, we conduct comprehensive ablation studies across multiple dimensions of LAGUNA: 1) methodological component analysis, 2) language model assessment for reference structure generation, 3) qualitative comparison of relative vs. absolute alignment in LAGUNA's features, 4) comparison with zero-shot MLLMs, 5) caption quality impact analysis, and 6) universal domain adaptation performance evaluation. Additional complementary ablation studies are provided in the supplementary material.

**Methodological Elements.** Table 4, ablates the main methodological elements of LAGUNA on GeoImnet. We set up this ablation by removing LAGUNA's core elements, namely the structure loss ($\mathcal{L}_S$ - Eq. (5)), domain-agnostic reference anchors $\mathcal{A}$, learnable anchors $\mathcal{A}_{t/s}$, cross-domain attention (CD Attn. - Eq. (8), (9)), and the regularization loss ($\mathcal{L}_{Reg}$ - Eq. (11)). We then progressively add these elements (settings (1)-(5)) until we reach LAGUNA. For all settings, we report the relative improvement (w.r.t. previous row) and the absolute one (w.r.t. first row).
*Absolute alignment* In setting (1), the visual classifier is trained on source and target samples using

| Setting | $\mathcal{L}_S$ | $\mathcal{A}$ | $\mathcal{A}_{t/s}$ | CD Attn. | $\mathcal{L}_{Reg}$ | Avg. Acc. | Rel. Imp. | Abs. Imp. |
|---------|------|------|------|------|------|------|------|------|
| (1) | - | - | - | - | - | 63.21 | - | - |
| (2) | - | ✓ | - | - | - | 63.99 | +0.78 | +0.78 |
| (3) | ✓ | ✓ | ✓* | - | - | 65.22 | +1.32 | +2.10 |
| (4) | ✓ | ✓ | ✓ | - | - | 67.08 | +1.77 | +3.87 |
| (5) | ✓ | ✓ | ✓ | ✓ | - | 67.52 | +0.44 | +4.31 |
| LAGUNA | ✓ | ✓ | ✓ | ✓ | ✓ | 68.68 | +1.16 | +5.47 |

Table 4: Ablation on LAGUNA's methodological components. '*' means that the learnable anchors $\mathcal{A}_{t/s}$ are domain-independent (i.e., the same set of learnable anchors for target and source).

labels and pseudo-labels predicted by the text supervisor, which leads to a baseline model akin to LaGTran. In (2), we add $\mathcal{A}$ and train the visual classifier as in setting (1), but we also add a loss to align visual representations with the related class anchors in absolute coordinates through cosine similarity. Setting (2) improves (1) by +0.87, suggesting that imposing a reference structure beyond pseudo-labels is beneficial to performance and that absolute alignment is too restrictive.

*Relative alignment* In (3), we extend (2) with domain-independent learnable anchors (✓∗ in Table 4, indicating that weights of $\mathcal{A}_t$ and $\mathcal{A}_s$ are shared) and add the $\mathcal{L}_S$ loss (Eq. 5). The introduction of learnable anchors enables the relative alignment of the visual domain with $\mathcal{A}$, leading to gains of +1.32 w.r.t (2) and +2.10 over (1), highlighting the benefits of the proposed relative alignment.

*Domain-Specific Anchors* Adding domain-specific anchors in setting (4) allows source and target domains to focus on the individual characteristics of the respective domains, leading to a further improvement of +1.77 w.r.t (3) and a robust +3.87 w.r.t the baseline (1).

*Cross-Domain Attention* Adding cross-domain attention in setting (5) bridges the gap between target and source representations, leading to+0.44 w.r.t (4) and an overall improvement of +4.31 w.r.t (1).

*Regularization Loss* We achieve the final configuration in LAGUNA, where we add the regularization loss, which prevents collapse in learning domain-specific anchors and leads to a further improvement of +1.16 w.r.t (5) and an overall gain of +5.47 w.r.t (1). These improvements show the efficacy of LAGUNA and the benefits of aligning visual representations to a reference structure.

**Language models for reference structures.** Fig. 3 a) reports semantic similarity maps for 100 randomly selected classes form GeoImnet when three different language models are used in stage 1 (*SentenceTransformer*, CLIP, and BERT), together with the average accuracy obtained keeping stages 2-3 fixed. Notably, accuracy follows the ability of models to distinguish classes, with low off-diagonal values indicating more discriminative models. While we achieve best results with *SentenceTransformer*, LAGUNA is robust to different reference structures, maintaining SOTA results.

**Absolute vs Relative Alignment.** Figure 3 b) shows t-SNE plots and related Maximum Mean Discrepancy (MMD) scores for 1000 randomly selected samples from the GeoImnet validation set, where the source is USA. Notably, features are separated in absolute coordinates, while they blend together (low MMD) in relative ones thanks to $\mathcal{L}_S$ and domain-specific learnable anchors. This highlights that semantic alignment can be achieved without forcing representations into a shared space, allowing each domain to preserve its unique characteristics while adhering to a common structure.

**Comparison with 'Zero-Shot' MLLMs.** Despite recent advances in MLLMs with billions of parameters trained on web-scale data, our experiments reveal that LAGUNA significantly outperforms them in challenging domain adaptation scenarios. Testing on two benchmarks requiring fine-grained spatial and temporal discrimination (GeoNet and Ego2Exo, Tab 5&6), we compare against five SOTA 7-8B MLLMs. LAGUNA achieved +11.84% higher accuracy on GeoNet and +2.78% in

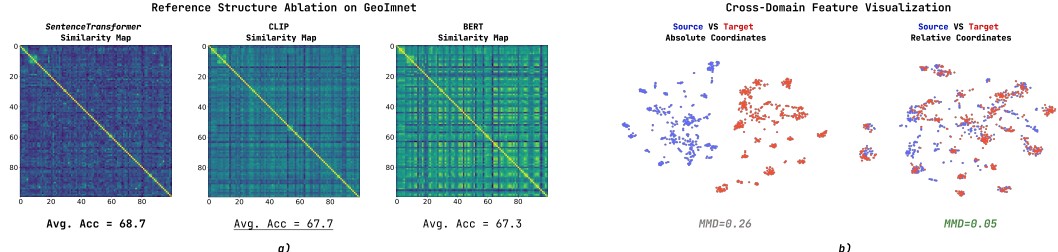

Figure 3: In a), similarity maps of 100 randomly selected classes from GeoImnet (yellow for high similarity) and average accuracies. In b), t-SNE plots for 1000 randomly selected *Source* and *Target* samples from GeoImnet, with respective MMD scores in Relative (right) and absolute (left) spaces.

| Model | GeoImnet | | GeoPlaces | | Avg. |
|---|---|---|---|---|---|
| | U→A | A→U | U→A | A→U | |
| LLaVA-Next 7B Li et al. (2024b) | 22.88 | 24.21 | 40.41 | 42.33 | 32.46 |
| LLaVA-OneVision 7B Li et al. (2024a) | 27.07 | 29.87 | 46.29 | 49.23 | 38.12 |
| Qwen2.5-VL 7B Bai et al. (2025) | 52.98 | 56.34 | 50.85 | 54.11 | 53.56 |
| InternVL3 8B Zhu et al. (2025) | 38.95 | 41.58 | 53.66 | 52.91 | 46.75 |
| PLM Cho et al. (2025) | 19.93 | 23.96 | 35.58 | 42.61 | 30.52 |
| LAGUNA | 67.39 | 69.97 | 61.15 | 63.51 | 65.40 |
| Improvement | +14.41 | +13.63 | +7.49 | +9.40 | +11.84 |

Table 5: Comparison on GeoImnet and GeoPlaces against zero-shot MLLMs.

| Model | Ego→Exo | Exo→Ego | Avg. |
|---|---|---|---|
| LLaVA-Next 7B Li et al. (2024b) | 4.28 | 4.95 | 4.62 |
| LLaVA-OneVision 7B Li et al. (2024a) | 5.57 | 10.18 | 7.88 |
| Qwen2.5-VL 7B Bai et al. (2025) | 11.46 | 20.39 | 15.93 |
| InternVL3 8B Zhu et al. (2025) | 13.07 | 28.34 | 20.71 |
| PLM Cho et al. (2025) | 22.54 | 37.16 | 29.85 |
| LAGUNA | 13.52 | 33.45 | 23.49 |
| Improvement | +0.45 | +5.11 | +2.78 |

Table 6: Comparison on Ego2Exo against zero-shot MLLMs. PLMs training set includes EgoExo-4D.

mean per-class accuracy on Ego2Exo. More importantly, in the Ego2Exo benchmark, we surpass InternVL-8B, which is trained in egocentric data and holds the state-of-the-art in many egocentric benchmarks. As a final remark, LAGUNA is approximately 100× smaller than these MLLMs, demonstrating that tailored models are still better at challenging UDA tasks.

| Model | S→C | C→S | Avg. |
|---|---|---|---|
| LLaVA-OneVision 7B Li et al. (2024a) | 82.20 | 74.10 | 78.15 |
| Qwen2.5-VL 7B Bai et al. (2025) | 82.35 | 74.26 | 78.31 |
| LAGUNA (LLaVA captions) | 82.51 | 76.20 | 79.35 |
| LAGUNA (Qwen2.5 captions) | 82.77 | 76.19 | 79.48 |

Table 7: Ablation on the quality of generated captions with different MLLMs on *sketch* and *clipart* domains from DomaiNet.

| Model | H-Score |
|---|---|
| Baseline Kalluri et al. (2024) | 50.20 |
| UniDA You et al. (2019) | 33.39 |
| DANCE Saito et al. (2020) | 51.75 |
| OVANet Saito and Saenko (2021) | 47.26 |
| LaGTran Kalluri et al. (2024) | 61.19 |
| **LAGUNA** | **64.41** |

Table 8: Experiment on GeoUniDA for universal domain adaptation.

**Caption Quality.** Table 1 presents LAGUNA results using BLIP-2-generated captions for consistency and fair comparison with previous works. In Table 7, we investigate how caption quality affects LAGUNA's performance by generating captions from two SOTA MLLMs: LLaVA-OneVision 7B and Qwen2.5-VL 7B, and evaluating LAGUNA on DomainNet's *sketch* and *clipart* domains, comparing its performance against the same MLLMs. Results show that LAGUNA benefits from higher-quality captioning models and offers an alternative that is 100x smaller while achieving better performance on specific domains. The supplementary material provides additional experiments demonstrating: 1) LAGUNA's robustness to low-quality captions and 2) its requirement for only a few target captions to achieve SOTA performance, reducing the one-time computational cost of automatic annotation with MLLMs. These findings make LAGUNA well-suited for real-world deployment in constrained environments where billion-parameter models are impractical.

**Universal Domain Adaptation.** Finally, we assess LAGUNA's performance in open-world domain transfer scenarios using the GeoUniDA dataset Kalluri et al. (2023), which, in contrast with traditional domain adaptation benchmarks, evaluates universal domain adaptation across domains, including both unique and shared semantic classes. Following OVANet Saito and Saenko (2021), we employ the H-score metric that balances closed-set and open-set accuracies through their harmonic mean. Notably, LAGUNA exhibits strong performance in this challenging setting, validating the effectiveness of our method.

## 5 CONCLUSIONS

We introduce LAGUNA, a novel domain adaptation approach leveraging geometrical structures of semantically equivariant spaces to guide adaptation. LAGUNA defines a reference representation space structure based on domain-agnostic class semantic similarities encoded by a language model, ensuring the organization of sample projections reflects this structure while preserving domain-specific characteristics. By conditioning the classifier to adhere to this structure, LAGUNA encourages structural similarity across domain-specific latent spaces, retaining unique features for improved classification. This is achieved through pseudo-labeling and learnable domain-specific anchors, guided by a loss function that prioritizes mimicking geometrical associations over direct representation alignment. Extensive experiments demonstrate LAGUNA's superior performance compared to existing methods, highlighting the importance of structural alignment and language-guided learning in domain adaptation. Future work could explore extending LAGUNA to multi-modal adaptation scenarios.

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

**Appendix LAGUNA: LAnguage Guided UNsupervised Adaptation with structured spaces**

Figure 4: LAGUNA's accuracy with different quantities of pseudo-labeled data (target samples with captions).

## A    INTRODUCTION

This supplementary material presents additional ablation studies complementing those in the main manuscript. We first examine the number of captions LAGUNA requires to achieve SOTA performance. Second, we extend caption quality experiments by testing LAGUNA's robustness with lower-quality captions. Third, we ablate language model choices and target domain structural guidance. Fourth, we provide detailed motivation for our regularization loss $\mathcal{L}_{Reg}$ in Eq. (8). Full code and implementation will be released upon acceptance. In addition to LAGUNA's code, we will also provide scripts to reproduce all the MLLMs results reported in this work.

## B    ABLATION ON THE RATIO OF TARGET SAMPLES.

Our approach relies on image captions, whether generated or sourced elsewhere. A potential concern with LAGUNA is that automatic annotation of large datasets can be tricky to obtain in full or even time-consuming. While the costs of obtaining these captions are minimal compared to manual annotation, we investigate the quantity of captions LAGUNA requires for SOTA performance in Fig. 4. We progressively increase randomly selected pseudo-labeled training data from 10% to 75% and observe the performance impact. Notably, LAGUNA achieves high accuracy with only 10% of samples and reaches SOTA results with just 20%. This efficiency highlights the benefits of our structure-driven approach, which organizes sample projections within domain-specific spaces according to a reference geometric structure.

## C    ABLATION ON CAPTIONING QUALITY.

LAGUNA achieves SOTA results with high-quality (EgoExo4D), generated (DomainNet), and low-quality (GeoNet) captions from web metadata, demonstrating high robustness. While the main manuscript showed that LAGUNA benefits from higher-quality captioning models like Qwen2.5-VL Bai et al. (2025) and LLaVA-OneVision Li et al. (2024a), real-world scenarios may involve suboptimal caption quality. We further examine LAGUNA's sensitivity to caption quality in Tab. 9 by comparing performance with BLIP-1 and BLIP-2 generated captions. Our main results (Tab. 1) use BLIP-2 captions from Kalluri et al. (2024), while BLIP-1 represents an earlier, less capable iteration with lower captioning quality. Transitioning from BLIP-2 to BLIP-1 results in only a

| Model | $S \to C$ | $C \to S$ |
|---|---|---|
| BLIP-1 Li et al. (2022) | 69.9 | 60.1 |
| BLIP-2 Li et al. (2023) | 72.0 | 64.1 |
| LAGUNA (BLIP-1) | 78.1 | 71.3 |
| LAGUNA (BLIP-2) | **80.2** | **73.5** |

Table 9: Ablation on caption quality with BLIP-1&2 captioning models. We consider sketch and clipart domains from DomainNet. The best result is in bold, and the second best is underlined.

| Scenario | *SentenceTransformer* | CLIP | BERT |
|---|---|---|---|
| U→A | 64.40 | 66.81 | 67.39 |
| A→U | 67.35 | 69.40 | 69.97 |
| **Avg.** | 65.87 | 68.11 | **68.68** |

Table 10: Ablation on three language models used in stage 2 and 3, *SentenceTransformer*, CLIP, and BERT with *SentenceTransformer*-defined $\mathcal{A}$ on GeoImnet with two domains: Asia (A) and Usa (U). The best result is in **bold**, and the second best is underlined.

2.2% performance reduction for C→S and 2.1% for S→C, confirming once again that LAGUNA benefits from caption quality but also that it is robust even in lower quality captions. Moreover, if we compare LAGUNA's results to the accuracy of classification obtained from BLIP-1&2 captions (through linear probing), we further emphasize the capability of LAGUNA to capture important signal from both the language (structure and caption pseudo-labels) and the visual space, which compensates for the quality of the captions.

## D ABLATION ON LANGUAGE MODELS

In addition to the language model used for defining the reference structure through $\mathcal{A}$ in Stage 1, LAGUNA employs language models in Stage 2 for encoding image/video captions trained for text classification and capturing semantic structure. This model is then used to generate pseudo-labels and serve as structure guidance for the target domain in Stage 3. We ablate the choice of language model in these Stages, comparing LAGUNA's performance when employing *SentenceTransformer* Reimers and Gurevych (2019), CLIP Radford et al. (2021), or BERT Sanh et al. (2019) on GeoImnet dataset in two settings: (1) using *SentenceTransformer*-defined $\mathcal{A}$ (Table 10) as it can better model semantic relationships (recall Fig. 3 from the main manuscript) and (2) defining $\mathcal{A}$ with the same language model as the one chosen for training (Table 11).

In setting (1), BERT outperforms CLIP and *SentenceTransformer*, motivating our choice for the language model. In setting (2), CLIP achieves the highest accuracy, but its overall performance remains lower than BERT's in setting (1). These results not only emphasize the best model and justify our choice but also demonstrate that using different models for structure definition and training can boost performance since the choices are tailored for their specific characteristics. Specifically, the Stage 1 model is selected for its ability to model meaningful relationships between classes, while Stage 2 is intended for building better representations of captions that learn the defined structure and can also produce better pseudo-labels.

## E ABLATION ON STRUCTURE GUIDANCE FOR TARGET DOMAIN IN STAGE-3

During Stage-3 training of LAGUNA, in addition to classification training, we train our model also for structure-preserving as defined from $\mathcal{A}$ using the structural loss $\mathcal{L}_S$. In this particular training loss, for the source domain, we supervise the visual relative representation $\mathbf{r}^{g_i^s} = rel(g_i^s, A_s)$ with the language relative encoding of the class corresponding language anchor $\mathbf{r}^{y_i^s} = rel(\mathcal{A}[y_i^s], \mathcal{A})$. On

| Scenario | *SentenceTransformer* | CLIP | BERT |
|----------|:---------------------:|:----:|:----:|
| U→A | 64.40 | 66.48 | 66.32 |
| A→U | 67.35 | 69.28 | 68.72 |
| **Avg.** | 65.87 | **67.88** | 67.52 |

Table 11: Ablation on three language models used in stage 2 and 3, *SentenceTransformer*, CLIP, and BERT, used for Stage-2 training on GeoImnet with two domains: Asia (A) and Usa (U). In this scenario, the same pre-trained model encodes $\mathcal{A}$ (stage 1). The best result is in **bold**, and the second best is underlined.

| | GeoImnet | | | | Ego2Exo | | | |
|---|:---:|:---:|:---:|:---:|:---:|:---:|:---:|:---:|
| | **U→A** | | **A→U** | | **Ego→Exo** | | **Exo→Ego** | |
| | $\mathbf{r}^{z_i^t}$ | $\mathbf{r}^{\overline{y}_i^t}$ | $\mathbf{r}^{z_i^t}$ | $\mathbf{r}^{\overline{y}_i^t}$ | $\mathbf{r}^{z_i^t}$ | $\mathbf{r}^{\overline{y}_i^t}$ | $\mathbf{r}^{z_i^t}$ | $\mathbf{r}^{\overline{y}_i^t}$ |
| | **67.39** | 66.77 | **69.97** | 68.14 | **13.52** | 13.49 | **33.45** | 32.88 |

Table 12: Ablation on structure guidance representations on GeoImnet and Ego2Exo. We consider the comparison between relative representations obtained from using the language encoding $\mathbf{r}^{z_i^t}$ and from using $\mathbf{r}^{\overline{y}_i^t}$. The best result for each scenario is in **bold**. Note that for Ego2Exo the reported metric is mean per class accuracy.

the other hand, in the target domain, since we do not have a ground truth label but only the estimated pseudo label $\overline{y}_i^t$, we supervise the visual relative representation $\mathbf{r}^{g_i^t} = rel(g_i^t, A_t)$ with the language relative encoding of the sample captioning $\mathbf{r}^{z_i^t} = rel(z_i^t, \mathcal{A})$ to benefit from the richer information expressed in the language encoding compared to the estimated pseudo-label. In this ablation, we investigate the benefits of using the $\mathbf{r}^{z_i^t}$ compared to relying on the pseudo-label and supervise the visual encoding with $\mathbf{r}^{\overline{y}_i^t} = rel(\mathcal{A}[y_i^t], \mathcal{A})$. Specifically, we ablate on the GeoImnet and Ego2Exo datasets and report the results in Table 12. Notably, $\mathbf{r}^{z_i^t}$ constantly gives better performances than $\mathbf{r}^{\overline{y}_i^t}$ in all scenarios motivating our choice to use the textual encoding rather than the pseudo-label corresponding anchor to create the relative representations for structure supervision in the target domain.

# F  REGULARIZATION LOSS EXTENDED

In Section 3.4.1 of the main manuscript, we explain how the structure loss $\mathcal{L}_S$ is supported by a regularization loss $\mathcal{L}_{Reg}$ to avoid feature collapsing. The logic behind $\mathcal{L}_{Reg}$ is built upon two base concepts of linear algebra: Gram matrix and determinant. To better explain this logic and its relation with volume and feature collapsing, in this section, we give more information about what is a Gram matrix, what its determinant represents, how it is connected to volume, and why it serves as a collapsing indicator to motivate our approach.

**Gram matrix.** Given a set of vectors $v_1, v_2, \ldots, v_n$ in $\mathbb{R}^d$, the Gram matrix $\gamma$ is defined as:

$$\gamma = V^T V$$

where $V$ is the matrix whose columns are the vectors $v_i$, and $\gamma$ is an $n \times n$ symmetric matrix whose elements are the pairwise inner products of the vectors:

$$\gamma_{ij} = \langle v_i, v_j \rangle$$

Additionally, its rank corresponds to the number of linearly independent vectors, and its determinant (if full-rank) relates to the volume of the parallelepiped spanned by the vectors. If $\gamma$ is not full rank, the vectors used to construct it are linearly dependent, so they do not span the full n-dimensional space. Geometrically, this implies that the volume of the parallelepiped they define is zero, as they

lie in a lower-dimensional subspace. In other words, a lower rank or a volume of zero of the vectors composing $\gamma$ indicates collapsing in a lower subspace. In LAGUNA, we use these properties of the Gram matrix to control and prevent the collapsing of the anchor vectors composing $\mathcal{A}_s$ and $\mathcal{A}_t$. Particularly, from Eq. (7) in the manuscript, we calculate $\gamma_s = A_s^T A_s$ and $\gamma_s = A_t^T A_t$.

**Determinant of Gram matrix.** To determine if a set of vectors has collapsed, one can either compute the rank of its Gram matrix or evaluate its volume in space. In LAGUNA, we choose the latter approach as we can compare it with a reference volume that helps prevent collapse during training. Specifically, to calculate the volume of the parallelepiped formed by the set of vectors (anchors) in $\mathcal{A}_{s/t}$, we utilize the determinant. The determinant of a Gram matrix represents the square of the volume of the parallelepiped defined by the vectors used to compute the matrix (e.g., $Det(\gamma_s)$ gives the square of the volume occupied by all anchors in $\mathcal{A}_s$). Thus, the volume for $\mathcal{A}_{s/t}$ can be expressed as:

$$\text{Volume}_{s/t} = \sqrt{Det(\gamma_{s/t})}.$$

Importantly, if the anchors in $\mathcal{A}_{s/t}$ are linearly dependent (i.e., not full rank), $Det(\gamma_{s/t}) = 0$. To prevent collapse, we can regularize $\mathcal{L}_S$ by requiring the volume to remain greater than zero. However, in higher-dimensional spaces, volumes can have very large values. For numerical stability, we replace $Det(\gamma_{s/t})$ with $logDet(\gamma_{s/t})$ in Eq. (8) and enforce that the space occupied by anchors in $\mathcal{A}_{s/t}$ remains log comparable to that of the reference anchors in $\mathcal{A}$ preventing their collapse into a zero volume. Specifically, Eq. (8) in the manuscript is defined as follows:

$$\begin{aligned}
\mathcal{L}_{Reg} \quad = \quad & |logDet(\gamma_t) - logDet(\gamma)| + \\
& |logDet(\gamma_s) - logDet(\gamma)|,
\end{aligned} \tag{11}$$

where $\gamma = \mathcal{A}^T \mathcal{A}$.

# G   USE OF GENERATIVE AI

According to the ICLR 2026 policy regarding the transparency of generative AI use in submitted works, we declare that generative AI tools were employed solely for language enhancement and correction purposes in this manuscript. All content, including technical contributions, experimental design, analysis, and conclusions, represents original work by the authors. We take full responsibility for every word reported in this paper, as each statement has been carefully reviewed, verified, and deliberately selected to accurately represent our research findings and contributions. The use of AI assistance was limited exclusively to improving grammatical clarity and linguistic precision, without influencing the scientific content or integrity of our work.

