# OpenReview forum: "LAGUNA: LAnguage Guided UNsupervised Adaptation with structured spaces"
_ICLR.cc/2026/Conference — Submitted to ICLR 2026_

### Official Review · Reviewer_Tqcr · 2025-10-26

**Soundness:** 3
**Presentation:** 3
**Contribution:** 3
**Rating:** 6
**Confidence:** 4

**Summary:**

The paper introduces LAGUNA (LAnguage-GUided domAiN Alignment), a framework for cross-domain visual recognition that aligns representations through language-defined semantic geometry. It first constructs a fixed language anchor space from class names, then trains a language supervisor to produce pseudo-labels and maintain this structure. In the final stage, domain-specific visual anchors and a cross-domain attention mechanism align visual features with the language geometry, while a volume regularization term preserves anchor diversity.

**Strengths:**

Using language both as a semantic reference space and as a source of pseudo-supervision offers an elegant way to bridge domains and modalities in the absence of target labels.

**Weaknesses:**

1. While the paper avoids reliance on large-scale vision-language models at inference, it does not adequately compare against methods that do use such models (e.g., CLIP, ALIGN, or recent prompt-tuning approaches in zero-shot or few-shot settings).
2. The three-stage pipeline introduces substantial training overhead. While each component is well motivated, the cumulative complexity may hinder adoption, particularly in settings with limited computational resources or high domain variability.

**Questions:**

1. The method relies on relative encodings (cosine similarities to semantic anchors) instead of directly aligning embeddings. Why is this preferable in practice? Did the authors compare it to direct matching of latent vectors (e.g., using contrastive or prototypical alignment)?
2. The class-name-based anchors A are fixed and assumed to encode reliable semantic geometry. Given that sentence embeddings may not reflect task-specific distinctions (e.g., action granularity), how sensitive is the framework to inaccuracies in these language embeddings?
3. The use of log-determinants of Gram matrices is elegant but potentially unstable or sensitive in high-dimensional spaces. Are there numerical issues during training? How does performance vary with or without this regularization term? Additionally,  can the authors provide intuition or theoretical reasoning for why this specific form of volume preservation is optimal over, say, minimizing pairwise collapse or encouraging orthogonality?
4. The method relies on aligning relative encodings rather than absolute feature vectors. Is there a formal justification or theoretical advantage to this approach in terms of generalization, invariance, or robustness across domains?
5. Given that multiple objectives (structure-preserving loss, cross-entropy, volume regularization) are optimized jointly across language and visual modalities, how are gradient conflicts handled? Was gradient interference an issue during training?

---

> ### Author Response · Authors · 2025-11-26
> **Question #1 - Relative vs Absolute Encoding**
>
> Thank you for this insightful question regarding the comparison between relative encodings and direct latent vector matching. We would like to point the reviewer to **Table 4** in Section 4.3 (Ablation Study), where we explicitly compare absolute alignment (direct matching) versus our proposed relative alignment approach. Specifically, the comparison can be seen in settings (1), (2), and (3):
>
> - **Setting (1)** represents a baseline where the visual classifier is trained using labels and pseudo-labels without any structural guidance, achieving 63.21% average accuracy on GeoImnet.
>
> - **Setting (2)** introduces absolute alignment by adding the reference anchors **A** and training with a loss that directly aligns visual representations to their corresponding class anchors in absolute coordinates through cosine similarity. This prototypical/direct matching approach yields only a modest improvement of **+0.78%** (63.99% accuracy).
>
> - **Setting (3)** introduces our relative alignment approach by adding the structural loss **L_S** (Eq. 5) with learnable anchors. This achieves **+1.32%** improvement over setting (2) and **+2.10%** over setting (1), reaching 65.22% accuracy.
>
> The additional gain of **+1.32%** when moving from absolute to relative alignment demonstrates the practical benefits of our approach. Furthermore, to provide additional evidence, we conducted experiments on DomainNet comparing absolute and relative alignment on the Clipart (C) and Sketch (S) domains. For the absolute alignment, we just substitute **L_S** with a direct cosine similarity between representations and anchors. Results are reported in Table 1 below, showcasing an improvement of **+2.17%** when using relative alignment rather than absolute one.
>
> **Table 1:** Comparison of absolute vs. relative alignment on DomainNet (Clipart and Sketch domains) using ViT-B backbone. Best results are in **bold**.
>
> | **Method** | **C→S** | **S→C** | **Avg.** |
> |:---|:---:|:---:|:---:|
> | LAGUNA (Absolute) | 71.14 | 78.3 | 74.72 |
> | LAGUNA (Relative) | **73.54** | **80.24** | **76.89** |
> | Improvement | **+2.44** | **+1.94** | **+2.17** |
>
> These results further confirm that relative alignment consistently outperforms direct matching across different datasets and domain pairs. This finding aligns with our motivation illustrated in Figure 1 of main manuscript: semantic equivalence can be achieved without forcing representations to overlap in absolute coordinates.

---

> > ### Author Response · Authors · 2025-11-26
> > **Question #2 - Descriptive anchors**
> >
> > This is a very interesting observation. Our goal when defining **A** (reference latent space) was to maintain the structural space as simple and domain-invariant as possible to allow for flexibility in the learned representations. One of the reasons for this design choice is that the datasets used in our evaluation often employ high-level class categories that encapsulate multiple fine-grained classes. For instance, in Ego2Exo, the class annotations from EgoExo4D group several specific fine-grained actions under broader category labels. This kind of hierarchical class composition is also found on GeoNet, making it challenging to represent them with a single detailed description that accurately captures all its fine-grained variations.
> >
> > However, to investigate the sensitivity of our framework to richer semantic anchors, we conducted experiments on the Ego2Exo benchmark, where describing actions might be more meaningful than using simple class definitions. We generated detailed action descriptions using Qwen 2.5-VL and used these descriptions to define the reference anchors **A** instead of class names. Results are reported in Table 2 below.
> >
> > **Table 2:** Comparison of class names vs. action descriptions as semantic anchors on Ego2Exo. We report mean per-class accuracy.
> >
> > | **Anchor Type** | **Ego→Exo** | **Exo→Ego** | **Avg.** |
> > |:---|:---:|:---:|:---:|
> > | Class Names | **13.52** | **33.45** | **23.49** |
> > | Action Descriptions | 12.25 | 32.51 | 22.38 |
> > | Δ | -1.27 | -0.94 | ≈-1.1 |
> >
> > Interestingly, the results show a performance decrease of approximately 1.1% when using detailed action descriptions. One particular observation that might explain this decrease is that the dense descriptions caused an increase in average cosine similarity between anchors, going from **0.265** with class names to **0.41** with action descriptions. This increased similarity is likely due to language redundancy in the descriptions, which reduces the discriminative power of the reference structure. This finding suggests that while richer semantic descriptions might seem beneficial, simpler class names can actually provide a more discriminative structural space for domain adaptation. However, we acknowledge that the dataset composition in our benchmarks is not the perfect setting for using rich semantic description.

---

> > > ### Author Response · Authors · 2025-11-26
> > > **Question #3 - Volumetric loss motivation**
> > >
> > > Thank you for this thoughtful question regarding the volume regularization term and its alternatives. We address each aspect of your inquiry below.
> > >
> > > **Motivation for Volumetric Loss.** The use of the log-determinant of Gram matrices was intended to encourage the learnable anchors **A_s** and **A_t** to not collapse into a very small region of the representation space. Since the anchors are used to compose the representations fed to our classifier (via the Cross-Domain Attention mechanism), maintaining meaningful distances between them is crucial for designing effective decision boundaries. The volumetric perspective naturally captures this requirement: if anchors collapse, the volume of the parallelepiped they span approaches zero, directly penalized by our loss.
> > >
> > > **Numerical Stability Considerations.** We acknowledge that Gram matrices are indeed sensitive in high-dimensional spaces. The most challenging aspect was finding the right balance of hyperparameters and transformation layers (i.e., linear vs. non-linear) when processing the anchors. Any imbalance could violate the constraint that the vectors composing the Gram matrix need to be strictly linearly independent, which proved more challenging to handle than initially anticipated. Our solution was to avoid MLPs in the initial transformation of anchors, combined with a learning rate warmup strategy. This approach ensured that the linear independence constraint was preserved throughout training and prevented numerical instabilities in the log-determinant computation. We will add this clarification in the revised manuscript.
> > >
> > > **Experimental Validation.** Regarding the impact of the regularization term, we refer the reviewer to **Table 4** in Section 4.3, where we compare LAGUNA with and without the volumetric loss **L_Reg**. Specifically, comparing setting (5) (without **L_Reg**) to the full LAGUNA configuration shows an improvement of **+1.16%** when including the regularization, demonstrating its importance in preserving better decision boundaries and improving classification performance.
> > >
> > > **Comparison with Pairwise Alignment.** We thank the reviewer for the suggestion regarding alternative regularization strategies. To investigate this, we conducted an experiment using direct Gram matrix alignment instead of their determinants, which effectively aligns pairwise similarities between anchors. This approach is similar to the one use by DINOv3 for feature diversity. Results on GeoImnet are reported in Table 3 below.
> > >
> > > **Table 3:** Comparison of volumetric loss vs. pairwise Gram matrix alignment on GeoImnet using ViT-B backbone.
> > >
> > > | **Regularization Type** | **U→A** | **A→U** | **Avg.** |
> > > |:---|:---:|:---:|:---:|
> > > | Volumetric (Log-Det) | **67.39** | **69.97** | **68.68** |
> > > | Pairwise (Gram Alignment) | 67.25 | 69.91 | 68.58 |
> > > | Δ | -0.14 | -0.06 | ≈-0.1 |
> > >
> > > Interestingly, the pairwise alignment approach shows only a slight performance drop of approximately 0.1% compared to the volumetric loss which can be probably avoided by adapting the hyperparameters to the new loss. However, we observed that training with pairwise alignment is notably more stable, as it is not constrained by the strict linear independence requirement of the Gram matrix determinant. We thank the reviewer for this valuable suggestion and will experiment further to evaluate the trade-off across all scenarios, given that pairwise alignment provides a more stable training procedure while achieving competitive performance.

---

> > > > ### Author Response · Authors · 2025-11-26
> > > > **Question #4 - Formal Justification for relative encoding**
> > > >
> > > > **Information-Theoretic Advantage: Preserving Sufficient Statistics**
> > > >
> > > > From an information-theoretic standpoint, absolute alignment enforces the constraint I(Z; D) → 0, requiring the representation Z to be independent of the domain D.
> > > >
> > > > **The Problem:** Theoretical work [1, 2] demonstrates a fundamental trade-off: if the domain D provides information about the label Y (e.g., under label shift where P_s(Y) ≠ P_t(Y), or when domain-specific "style" aids recognition), then I(Y; D) > 0. Zhao et al. [1] prove an information-theoretic lower bound showing that when marginal label distributions differ between domains, forcing domain invariance while minimizing source error will *only increase* the target error, a phenomenon they characterize as an "uncertainty principle" for domain adaptation. Johansson et al. [2] further show that domain invariance is "often a far too strict requirement" that causes information loss through non-invertible transformations.
> > > >
> > > > **Our Solution:** LAGUNA does not force I(Z; D) → 0. By utilizing distinct domain-specific anchors **A_s** ≠ **A_t**, we allow the encoding to retain domain-specific information (maximizing entropy H(Z)). Simultaneously, the structural loss **L_S** ensures that the *conditional entropy* of the label given the relative structure is minimized (H(Y | rel(Z)) → 0). This achieves a theoretically superior trade-off: maintaining domain-specific informativeness (sufficient statistics) while ensuring semantic alignment via structure.
> > > >
> > > > We will make sure to insert such theoretical motivation in our revised manuscript.
> > > >
> > > > [1] Zhao et al., "On Learning Invariant Representations for Domain Adaptation", ICML 2019.
> > > > [2] Johansson et al., "Support and Invertibility in Domain-Invariant Representations", AISTATS 2019.

---

> > > > > ### Author Response · Authors · 2025-11-26
> > > > > **Question #5 - Gradient Stability**
> > > > >
> > > > > Thank you for this question regarding gradient dynamics during training. Gradient conflicts were not a significant issue in our experiments. The multiple objectives (structure-preserving loss **L_S**, cross-entropy loss **L_CE**, and volume regularization **L_Reg**) co-exist harmoniously during optimization, as they address complementary aspects of the learning problem: **L_CE** ensures discriminative classification, **L_S** enforces structural alignment to the reference space, and **L_Reg** prevents anchor collapse.
> > > > >
> > > > > The primary challenge we encountered, as discussed in our response regarding the regularization loss, was ensuring that the linear independence constraint on the anchor vectors was preserved throughout training. This was addressed through architectural choices (avoiding MLPs in the initial anchor transformations) and training strategies (learning rate warmup), rather than explicit gradient conflict resolution mechanisms.
> > > > >
> > > > > To quantify training stability, we monitored gradient norms throughout our experiments. The average gradient norm during training remains below 1.0 × 10⁻¹, indicating stable optimization dynamics without the need for gradient clipping or specialized multi-objective optimization techniques. This stability can be attributed to the careful calibration of loss weights (λ₁, λ₂, λ₃) that normalize the magnitudes of each loss component, as described in Section 4.1 of our manuscript.
> > > > >
> > > > > In summary, the joint optimization of our multiple objectives proceeds smoothly without requiring explicit gradient conflict handling, which we attribute to the complementary nature of our loss terms and the appropriate scaling of their contributions.

---

> > > > > > ### Author Response · Authors · 2025-11-26
> > > > > > **Weakness #1 - Zero-shot comparison**
> > > > > >
> > > > > > We thank the reviewer for this comment and the opportunity to clarify our methodological positioning.
> > > > > >
> > > > > > **Methodological distinction:** LAGUNA is fundamentally designed for *unsupervised domain adaptation*, zero-shot transfer is not part of our model capabilities. Our approach learns domain-specific representations aligned to a language-guided reference structure, rather than relying on the extended world knowledge embedded in large-scale vision-language models, which unlocks the zero-shot adaptation. This is a deliberate design choice; we aim to demonstrate that structural alignment between domains can be achieved through learned representations, which is beneficial to absolute alignment. Such a representation learning strategy is difficult to apply to the existing VLM backbones as it goes against their contrastive learning approach.
> > > > > >
> > > > > > **Existing comparisons:** That said, we do provide comparisons against VLM-based approaches to contextualize our results:
> > > > > > - **Table 1 (ViT-B):** CLIP zero-shot, DAPrompt, PADCLIP, and UniMoS—all leveraging CLIP pretraining or prompt-tuning
> > > > > >
> > > > > > **Future directions:** We acknowledge the potential of combining LAGUNA's relative alignment framework with large-scale VLMs to enable zero-shot capabilities. Exploring how our structure-preserving approach could complement the rich semantic knowledge of such models is an exciting direction we plan to pursue in future work.

---

> > > > > > > ### Author Response · Authors · 2025-11-26
> > > > > > > **Weakness #2 - Computational overhead**
> > > > > > >
> > > > > > > We appreciate the reviewer's concern regarding computational efficiency. We clarify that caption generation is a **one-time offline process**, and the runtime overhead of LAGUNA is negligible compared to standard UDA baselines. We provide a detailed efficiency analysis below:
> > > > > > >
> > > > > > > * **Offline Caption Generation Cost:**
> > > > > > >     Caption generation is performed only once per dataset. Using a V100 GPU and 8-bit quantization, a modern captioning model like BLIP-2 (OPT-2.7B) has an inference latency of approximately $\\sim 0.4$ seconds per image (non-batched). For a large-scale dataset of $50$K images, the total generation time is $\\sim 6$ GPU hours. This one-off cost is minor compared to the cumulative time required for multi-epoch training and hyperparameter tuning, which is a standard in deep learning. In our practical case, we used BLIP-2 captions provided by LaGTran [1]. However, this is the cost analysis for a V100 GPU, which should be dramatically lower if optimized with batched processing.
> > > > > > >
> > > > > > > * **Training Efficiency:**
> > > > > > >     LAGUNA's training pipeline is highly efficient relative to the baseline. On a V100 GPU:
> > > > > > >     * **Stage 1 (Reference Structure):** Requires no training; embedding 600 class labels takes $\\sim 1$ minute.
> > > > > > >     * **Stage 2 (Language Supervisor):** Training the text supervisor requires only $\\sim 9$ GPU-minutes per epoch.
> > > > > > >     * **Stage 3 (Visual Classifier):** The additional computation adds $<2$ GPU-minutes per epoch overhead. Considering the baseline training time of $\\sim 65$ GPU minutes per epoch, this represents a negligible increase of $\\sim 3\\%$. The main cause of this increase is caption encoding during training to obtain $z_t^i$, which is not needed in inference.
> > > > > > >
> > > > > > > * **Inference Efficiency:**
> > > > > > >     At test time, LAGUNA utilizes a ViT-B backbone ($\\sim 86$M parameters, $17.6$ GFLOPs). Our method adds only $\\sim 5$M parameters (via learned anchors, lightweight cross-attention, and MLP). Since the cross-attention operates on global feature tokens (1 global token per image $g_i$) rather than the entire sequence of patches, the additional floating-point operations (FLOPs) are negligible ($\\sim 17.61$ GFLOPs). Consequently, LAGUNA's inference speed is almost identical to the standard backbone. (FLOPs are calculated using the fvcore library).
> > > > > > >
> > > > > > > In summary, LAGUNA achieves superior performance with minimal computational overhead, ensuring high practicality for real-world deployment. We will make sure to include this analysis of LAGUNA's computational costs in the final version of our work.

---

> ### Author Response · Authors · 2025-11-26
>
> Dear Reviewer Tqcr,
>
> We sincerely thank you for your time and valuable feedback on our submission. In the following response, we have addressed your concerns to the best of our abilities.
>
> We would greatly appreciate any further feedback you may have, and we remain happy to address any additional questions or concerns that may arise.
>
> Best regards,
> The Authors

---

### Official Review · Reviewer_B8Zf · 2025-10-26

**Soundness:** 2
**Presentation:** 3
**Contribution:** 2
**Rating:** 4
**Confidence:** 4

**Summary:**

This paper proposes a language-guided approach to address the unsupervised domain adaptation (UDA) problem by aligning source and target domains in relative spaces. It first highlights the advantages of relative representations over absolute alignment. Then, it introduces a method that aligns source and target feature spaces with a language-based reference structure to achieve domain-invariant yet discriminative representations. Experimental results across 4 benchmarks demonstrate the effectiveness of the proposed method.

**Strengths:**

1. The investigation comparing absolute and relative alignment (fig. 1) is interesting. It shows that domain-specific characteristics such as color or texture, though differing across domains, can still be useful for classification within each domain.

2. The idea of using language to construct a reference structure is interesting as language provides a semantic space agnostic to visual variations.

3. The paper is clearly written and easy to follow.

**Weaknesses:**

1. While the paper argues that it is important to preserve domain-specific representations in the aligned space and provides a two-point example (lines 52–70), I am still concerned that such domain-specific representations inherently reflect domain shift and should ideally be minimized. Could you do some experiments to justify and empirically support your motivation.

2. In fig. 2, it would be helpful to indicate which components are learnable using visual icons, rather than only marking the frozen parts. Additionally, in Stage 3, $\mathbf{r}_i^{y^s}$ appears to be mistakenly written as $\mathbf{r}_i^{y^t}$. Moreover, this figure (on page 3) can only be fully understood when read together with the corresponding explanation on page 4,5, since readers may not immediately know what $\mathbf{r}_i^{g^t}$, $\mathbf{r}_i^{g^s}$, $\mathbf{r}_i^{y^s}$ and $\mathbf{r}_i^{z^t}$represent.

3. The paper does not clearly describe how the domain-specific anchors $A_t$ and $A_s$ are initialized. Could you please explicitly state their initialization strategy and further clarify why these anchors need to be learnable rather than fixed.

4. Since the proposed method involves caption generation for both source and target domains, it may introduce additional computational overhead compared to traditional UDA approaches. Could you provide a time or efficiency analysis (e.g., training and inference time, caption generation cost) to demonstrate the practicality of your method relative to existing UDA baselines.

5. Regarding the benchmark evaluation, the paper does not include comparisons with more recent methods published this year, even though it is already October. For example, I found a recent work [1] that also uses language guidance to address the UDA problem. It would strengthen the evaluation if the authors could include this method in their benchmark for a fairer and more up-to-date comparison.

6. In line 390, there is a typo: the improvement value should be 0.78 instead of 0.87.

[1] Litrico, Mattia, et al. "TRUST: Leveraging Text Robustness for Unsupervised Domain Adaptation." arXiv preprint arXiv:2508.06452 (2025).

**Questions:**

Please see the above weaknesses.

---

> ### Author Response · Authors · 2025-11-20
> **Response to Reviewer: Weakness #1 - 3**
>
> **Weakness #1 - Need for Relative Alignment**
>
> We thank the reviewer for the valuable comment. While minimizing domain shift is standard in UDA, our experiments demonstrate that enforcing strict absolute alignment discards discriminative cues necessary for classification. We empirically justify our motivation using Table 4 in the manuscript. Specifically, we conduct the experiments on Geoimnet USA $\Rightarrow$ Asia and Asia $\Rightarrow$ USA, which can be identified as Setting (2) and Setting (4). The results are the following:
>
> * Setting (2) enforces absolute alignment (minimizing domain-specific variations as suggested), resulting in 63.99% accuracy.
> * Setting (4) enables domain-specific representations via separate learnable anchors ($\\mathcal{A}_s, \\mathcal{A}_t$), improving accuracy to 67.08% (+3.09%).
>
> This significant gain confirms that preserving domain-specific features enhances adaptation performance. Furthermore, Figure 3b visually supports this: while source and target representations remain distinct in absolute coordinates (preserving domain characteristics), they achieve near-perfect alignment in relative coordinates (MMD: 0.005), effectively resolving the domain shift without suppressing domain-specific information. We will ensure that this point is emphasized more clearly in our manuscript.
>
> **Weakness #2 - Figure Adjustment**
>
> We thank the reviewer for pointing this out, we will update our figur accordingly to make the comprehension easier.
>
> **Weakness #3 - Anchor Initialization and Learnability**
>
> We thank the reviewer for bringing this omission to our attention. We acknowledge that the initialization strategy was not explicitly detailed in the main text and will ensure it is clarified in the final version. To answer your question: the domain-specific anchors $\mathcal{A}_s$ and $\mathcal{A}_t$ are randomly initialized for empirical purposes. They must be learnable to allow the source and target spaces to adapt to their specific distributions while maintaining structural alignment. The empirical results (shown in Table 1 & 2) are conducted on GeoImnet, consistent with our ablations in the main paper:
>
> 1. **Fixed vs. Learnable:** Fixing the anchors (making them non-learnable) results in a performance drop of approximately $5\\%$, demonstrating that flexibility is crucial for capturing domain-specific nuances. This experiment is also present in the manuscript, Table 4 setting (2) and LAGUNA.
>
> 2. **Initialization Strategy:** We experimented with initializing anchors using *mean visual sample encodings* or *reference anchors*. For the former, we initialize the anchors by averaging 100 randomly selected visual samples for each class from the source set. Both strategies yielded slightly lower results compared to random initialization, confirming that learning from a random start allows the model to find a more optimal configuration for the specific domain distributions.
>
> **Table 1: Ablation study on the learnability of domain-specific anchors on GeoImnet.**
>
> | Configuration | U $\to$ A | A $\to$ U | Avg. |
> | :--- | :---: | :---: | :---: |
> | Fixed Anchors (init from $A$) | 61.50 | 66.48 | 63.99 |
> | Learnable Anchors (Ours) | **67.39** | **69.97** | **68.68** |
>
> **Table 2: Impact of different anchor initialization strategies on GeoImnet performance.**
>
> | Initialization Strategy | U $\to$ A | A $\to$ U | Avg. |
> | :--- | :---: | :---: | :---: |
> | Reference Anchors | 67.01 | 68.56 | 67.79 |
> | Mean Visual Encoding | 67.32 | 69.70 | 68.51 |
> | Random | **67.39** | **69.97** | **68.68** |
>
> We will make sure to insert this ablation on the revised supplementary material.

---

> ### Author Response · Authors · 2025-11-20
> **Weakness #4**
>
> **Weakness #4 - Computational Cost**
>
> We appreciate the reviewer's concern regarding computational efficiency. We clarify that caption generation is a **one-time offline process**, and the runtime overhead of LAGUNA is negligible compared to standard UDA baselines. We provide a detailed efficiency analysis below:
>
> * **Offline Caption Generation Cost:**
>     Caption generation is performed only once per dataset. Using a V100 GPU and 8-bit quantization, a modern captioning model like BLIP-2 (OPT-2.7B) has an inference latency of approximately $\\sim 0.4$ seconds per image (non-batched). For a large-scale dataset of $50$K images, the total generation time is $\\sim 6$ GPU hours. This one-off cost is minor compared to the cumulative time required for multi-epoch training and hyperparameter tuning, which is a standard in deep learning. In our practical case, we used BLIP-2 captions provided by LaGTran [1]. However, this is the cost analysis for a V100 GPU, which should be dramatically lower if optimized with batched processing.
>
> * **Training Efficiency:**
>     LAGUNA's training pipeline is highly efficient relative to the baseline. On a V100 GPU:
>     * **Stage 1 (Reference Structure):** Requires no training; embedding 600 class labels takes $\\sim 1$ minute.
>     * **Stage 2 (Language Supervisor):** Training the text supervisor requires only $\\sim 9$ GPU-minutes per epoch.
>     * **Stage 3 (Visual Classifier):** The additional computation adds $<2$ GPU-minutes per epoch overhead. Considering the baseline training time of $\\sim 65$ GPU minutes per epoch, this represents a negligible increase of $\\sim 3\\%$. The main cause of this increase is caption encoding during training to obtain $z_t^i$, which is not needed in inference.
>
> * **Inference Efficiency:**
>     At test time, LAGUNA utilizes a ViT-B backbone ($\\sim 86$M parameters, $17.6$ GFLOPs). Our method adds only $\\sim 5$M parameters (via learned anchors, lightweight cross-attention, and MLP). Since the cross-attention operates on global feature tokens (1 global token per image $g_i$) rather than the entire sequence of patches, the additional floating-point operations (FLOPs) are negligible ($\\sim 17.61$ GFLOPs). Consequently, LAGUNA's inference speed is almost identical to the standard backbone. (FLOPs are calculated using the fvcore library).
>
> In summary, LAGUNA achieves superior performance with minimal computational overhead, ensuring high practicality for real-world deployment. We will make sure to include this analysis of LAGUNA's computational costs in the final version of our work.

---

> ### Author Response · Authors · 2025-11-20
> **Weakness #5 - 6**
>
> **Weakness #5 - Comparison with the recent work**
>
> Thank you for your constructive feedback; it is genuinely appreciated and will help strengthen our evaluation. We appreciate you bringing TRUST (Litrico et al., arXiv:2508.06452) to our attention. As this work was a preprint in August 2025, very close to the ICLR submission deadline, it is concurrent work we were not aware of. However, we agree that comparing it strengthens our evaluation.
>
> **Methodologically**, the approaches are distinct: TRUST utilizes contrastive learning with caption-guided soft labels to denoise pseudo-labels in a shared absolute space. In contrast, LAGUNA employs relative alignment, ensuring source and target domains share a geometric structure while maintaining their domain-specific absolute coordinates.
>
> **Empirically**, as shown in Table 1 & 2, LAGUNA compares favorably, outperforming TRUST on DomainNet (+0.78% Swin-B, +3.59% ViT-B), GeoPlaces (+2.22%), and GeoImnet (+2.3%). Moreover, LAGUNA demonstrates broader versatility, achieving SOTA results in video adaptation (EgoExo4D) and universal domain adaptation (GeoUniDA).
>
> **Table 1: DomainNet dataset results for domain adaptation. The best results are reported in **bold** while the second best are in *italics*.**
>
> | Model | Real$\\rightarrow$C | Real$\\rightarrow$S | Real$\\rightarrow$P | Clipart$\\rightarrow$R | Clipart$\\rightarrow$S | Clipart$\\rightarrow$P | Sketch$\\rightarrow$R | Sketch$\\rightarrow$C | Sketch$\\rightarrow$P | Paint$\\rightarrow$R | Paint$\rightarrow$C | Paint$\rightarrow$S | Avg. |
> | :--- | :---: | :---: | :---: | :---: | :---: | :---: | :---: | :---: | :---: | :---: | :---: | :---: | :---: |
> | TRUST (Swin-B) | **81.04** | *71.12* | 69.95 | 81.42 | 69.15 | 69.03 | 81.44 | 79.37 | *72.77* | 81.61 | *78.33* | 64.11 | 74.95 |
> | LAGUNA (Swin-B) | *80.34* | 70.68 | *71.92* | *83.07* | *69.51* | *70.59* | *83.34* | *79.71* | 70.51 | *83.32* | 77.47 | *68.32* | *75.73* |
> | LAGUNA (ViT-B) | 79.53 | **73.18** | **74.24** | **86.04** | **73.54** | **74.98** | **85.81** | **80.24** | **74.43** | **86.57** | **79.89** | **74.11** | **78.54** |
>
> **Table 2: Results on GeoImnet and GeoPlaces with 4 adaptation scenarios using ViT-B. Best results are in **bold**.**
>
> | Model | GeoImnet U$\\rightarrow$A | GeoImnet A$\\rightarrow$U | GeoPlaces U$\\rightarrow$A | GeoPlaces A$\\rightarrow$U | Avg. |
> | :--- | :---: | :---: | :---: | :---: | :---: |
> | TRUST | 65.77 | 67.02 | 59.89 | 60.14 | 63.20 |
> | LAGUNA | **67.39** | **69.97** | **61.15** | **63.51** | **65.40** |
>
> **Weakness #6 - Typo**
>
> We thank the reviewer and we will adjust the number.

---

> ### Author Response · Authors · 2025-11-26
>
> Dear Reviewer B8Zf,
>
> We sincerely thank you for your time and valuable feedback on our submission. In the following response, we have addressed your concerns to the best of our abilities.
>
> We would greatly appreciate any further feedback you may have, and we remain happy to address any additional questions or concerns that may arise.
>
> Best regards,
> The Authors

---

### Official Review · Reviewer_sRT2 · 2025-10-28

**Soundness:** 3
**Presentation:** 2
**Contribution:** 2
**Rating:** 2
**Confidence:** 4

**Summary:**

This paper presents LAGUNA (LAnguage Guided UNsupervised Adaptation), a new unsupervised domain adaptation framework that replaces traditional absolute feature alignment with concept-level relational alignment. Instead of forcing cross-domain samples to collapse into a shared latent space — which may destroy domain-specific structure — LAGUNA uses language to define a domain-agnostic semantic geometry over class labels, and then constrains visual features so that their inter-class relationships respect this reference geometry. By aligning the relative positioning of equivalent concepts rather than raw coordinates, LAGUNA preserves domain-specific characteristics while enforcing consistent semantics across domains, showing that semantic alignment can be achieved without projecting all domains into a single shared space.

**Strengths:**

This paper presents LAGUNA (LAnguage Guided UNsupervised Adaptation), a new unsupervised domain adaptation framework that replaces traditional absolute feature alignment with concept-level relational alignment. Instead of forcing cross-domain samples to collapse into a shared latent space — which may destroy domain-specific structure — LAGUNA uses language to define a domain-agnostic semantic geometry over class labels, and then constrains visual features so that their inter-class relationships respect this reference geometry. By aligning the relative positioning of equivalent concepts rather than raw coordinates, LAGUNA preserves domain-specific characteristics while enforcing consistent semantics across domains, showing that semantic alignment can be achieved without projecting all domains into a single shared space.


Through its architectural design and corresponding experiments, this paper demonstrates that semantic alignment can be achieved without enforcing representations from different domains to reside in a shared space, thereby allowing each domain to retain its unique characteristics while maintaining consistency with a common structural framework.

**Weaknesses:**

1. It is hard to say that this paper is a piece of work of UDA due to the usage of captions. Moreover, for low-quality captions, the ablation study in this paper only examines the effect of using a weaker caption generation model (i.e., replacing BLIP-2 with BLIP-1). However, it does not evaluate the robustness of the proposed method under more challenging cases, such as when captions omit the target category or include incorrect class information.

2. The key point of this paper is leveraging the conception space to bridge the source and target domains. Namely, the conception space is treated as a proxy space of the latent domain-invariant space. It is not a new idea. The differences from prior work are not clearly stated.

3. For Eq. (8)–(9), the paper lacks theoretical justification or empirical evidence explaining the rationale behind grounding gti and gsi to the structure of source anchors As instead of At. The underlying principle and the performance advantage of this design choice should be further clarified and supported.

4. In Section “3 METHOD,” the notation for the language-based, domain-agnostic reference structure A is inconsistent with the symbol *A used in Fig. 2, which may cause confusion in understanding the formulation and should be unified for clarity.

5. Given the stated goal “to ensure that source and target visual representations are grounded in the structure imposed by A, while still being free to capture domain-specific nuances,” it would be valuable to include experiments that explicitly analyze and visualize how As and At reflect this balance between adherence to the shared reference structure A and preservation of domain-specific characteristics.

**Questions:**

Refer to the weaknesses above.

---

> ### Author Response · Authors · 2025-11-20
> **Response to Reviewer: Weakness #1**
>
> We appreciate the reviewer's constructive feedback. We address the concerns regarding the definition of UDA and robustness to caption noise below.
>
> **1. Validity of Caption-Guided UDA.**
>
> We respectfully clarify that leveraging language signals (e.g., captions, VLM-generated pseudo-labels) is a well-established and growing paradigm in UDA. Our approach builds directly on this foundation.
>
> Specifically, several recent works have successfully incorporated language signals into UDA. For instance, [1] and [2] utilize language captions to generate pseudo-labels as core components of their UDA methodology. Similarly, [3] and [4] (along with other works cited in our manuscript) employ CLIP-based models to either generate pseudo-labels or utilize domain-specific prompts, leveraging CLIP's language capabilities as weak supervision signals for training UDA models. Our use of VLM-generated captions (provided by [1] and used also by [2]) follows this same principle, providing weak supervision through automatically generated language descriptions.
>
> We acknowledge that this represents an evolution from classical UDA that relies purely on vision. However, given that methods like [1-4] have been widely accepted in the UDA community and the captions are generated automatically without any manual annotation (in line with UDA settings), we believe our approach aligns well with current standards in the field. We would be happy to clarify this positioning in the paper if the reviewer believes additional discussion is needed.
>
> **2. Robustness to "Incorrect Class Information"**
>
> The reviewer correctly notes that the BLIP-1 vs. BLIP-2 ablation might not fully isolate failure modes like "incorrect classes." To address this, we conducted a granular semantic audit of BLIP-1 & 2 generated captions on DomainNet (Sketch and Clipart) and GeoNet (captions taken from web metadata) using Qwen 2.5-7B (prompt provided below, while captions can be found at [1]). We evaluated whether each caption: (a) explicitly contains the ground-truth class label or a semantic equivalent, and (b) correctly describes the category. With the decision that we get from the LLM, we compute a compatibility score, which is the number of captions correctly describing or containing the class divided by the total number of captions.
>
> Our analysis (Table 1) reveals that GeoImnet captions are significantly noisy, with a semantic compatibility score of $\le 50\\%$, indicating that half of the captions fail to correctly describe the class category in the USA domain. In contrast, BLIP-generated captions for DomainNet are more consistent: BLIP-1 achieves compatibility scores of $66\\%$ (Sketch) and $76.69\\%$ (Clipart), while BLIP-2 improves this to $73.64\\%$ and $82.04\\%$, respectively. Despite the extreme noise level in GeoImnet ($50\\%$ of the guidance is possibly incorrect), LAGUNA maintains robust performance (+$4.77\\%$ improvement compared to previous SOTA), demonstrating its resilience to the specific challenging conditions (omissions and incorrect class information) raised by the reviewer.
>
> **Table 1: Semantic Compatibility Analysis of Captions across Domains using Qwen 2.5-7B.**
>
> | Dataset | Domain | Compatibility Score (%) |
> | :--- | :--- | :---: |
> | GeoNet | USA | 49.99 |
> | GeoNet | Asia | 54.69 |
> | DomainNet | Sketch (BLIP-1) | 66.00 |
> | DomainNet | Clipart (BLIP-1) | 76.69 |
> | DomainNet | Sketch (BLIP-2) | 73.64 |
> | DomainNet | Clipart (BLIP-2) | 82.04 |
>
> **Prompt used for the analytics:**
>
> > You are evaluating whether an image caption correctly describes an object of a given class.
> >
> > Class Name: `{class_name}`
> >
> > Caption: `{caption}`
> >
> > Your task is to determine if the caption contains the class name or a synonym/related term that refers to the same object/entity.
> >
> > **Guidelines:**
> > - contains class or synonym: true if the caption mentions the class name or a synonym (e.g., "dog" for "canine", "car" for "automobile", "airplane" for "aircraft")
> > - correctly describes class: true if the caption's description is consistent with the class, even if it doesn't explicitly mention it (e.g., "a four-legged pet animal" for class "dog")
> > - Be strict: if the caption describes a different object or contradicts the class, answer with false, otherwise with true.
> >
> > DO NOT OUTPUT ANYTHING OTHER THAN THE TRUE OR FALSE ANSWER.
>
> **References**
>
> [1] Kalluri et al., Tell, don’t show!: Language guidance eases transfer across domains in images and videos, ICML 2024
>
> [2] Lirico et al., TRUST: Leveraging Text Robustness for Unsupervised Domain Adaptation, AAAI 2026
>
> [3] Ge et al., Domain Adaptation via Prompt Learning, IEEE Transactions on Neural Networks and Learning Systems (TNNLS) 2023
>
> [4] Du et al., Domain-Agnostic Mutual Prompting for Unsupervised Domain Adaptation, CVPR 2024

---

> ### Author Response · Authors · 2025-11-20
> **Weakness #2 - 4**
>
> **Weakness #2 - Contribution Novelty**
>
> We thank the reviewer for this opportunity to clarify our contribution. We believe there may be a misunderstanding regarding how we utilize the "conception space."
>
> We fully agree that using a semantic concept space (e.g., language embeddings) as a proxy for domain invariance is an established direction (e.g., CLIP-based methods, LaGTran). **However, we do not claim novelty in using language as a guide; rather, we claim novelty in *how* we align to it.**
>
> To clarify the distinction:
>
> * **Prior Work (Absolute Alignment):** Most existing language-guided UDA methods (e.g., [1], [2]) treat the concept space as a **destination**. They force source and target visual embeddings to map directly to the *same* absolute coordinates as their corresponding text embeddings (i.e., minimizing $\\|z_{img} - z_{text}\\|$). This forces the visual encoder to discard domain-specific information (e.g., texture, lighting) that does not exist in the text encoder.
>
> * **LAGUNA (Relative/Structural Alignment):** We treat the concept space as a **geometric blueprint**. We do not force visual embeddings to match text embeddings in absolute space. Instead, we enforce that the **inter-class relationships** (angles) in the visual space mirror the relationships in the concept space (i.e., matching the *topology* of the spaces).
>
> This allows LAGUNA to maintain **domain-specific absolute representations** (preserving discriminative domain cues) while achieving **semantic alignment via relative structure**. To our knowledge, this application of relative representation learning to UDA is novel.
>
> Since the review mentions that this is "not a new idea" without citing specific papers, we are concerned we may be missing specific works that use **relative/structural alignment** in UDA. We would be very grateful if the reviewer could provide these references so we can discuss them and position our work more accurately.
>
> **Weakness #3 - Motivation of Eq (8) - (9)**
>
> We thank the reviewer for this insightful observation. This indeed deserves clearer justification.
>
> The structural loss $L_S$ (Eq. 5) enforces that both domains maintain geometric relationships aligned to the reference structure $A$:
> $rel(g^{s_i}, A_s) \\simeq rel(A[y^{s_i}], A)$  and $rel(g^{t_i}, A_t) \\simeq rel(z^{t_i}, A)$.
>
> Since both $rel(A[y^{s_i}], A)$ and $rel(z^{t_i}, A)$ reference the same structure $A$, and both visual encodings are trained to mirror this structure through their respective anchors, the relative positioning of samples with respect to their domain-specific anchors becomes structurally equivalent across domains. This structural equivalence is crucial, as the attention patterns $QK^T$ computed in each domain capture semantically similar relationships, even though they use different keys ($A_t$ vs. $A_s$).
>
> Given this alignment, we can safely use $A_s$ as values for both domains, as the attention mechanism effectively utilizes domain-specific $Q \\cdot K^T$ patterns to select which anchors are relevant, then aggregates those anchors for the output. In addition, using $A_s$ as values for both domains offers significant practical advantages, as $A_s$ is trained with ground-truth labels, while $A_t$ relies on noisy pseudo-labels. To show how this benefits the results, we conducted an experiment on GeoImnet where each domain representation is grounded to its corresponding anchors in Table 1. As can be noticed, the results show an average improvement of +0.77 on this task, which motivates our design decision.
>
> **Table 1: Ablation on grounding cross-attention in Eq. (8)-(9) on domain-specific anchors.**
>
> | Config | GeoImnet U$\\rightarrow$A | GeoImnet A$\\rightarrow$U | Avg. |
> | :--- | :---: | :---: | :---: |
> | Domain-Specific Values | 66.79 | 69.03 | 67.91 |
> | LAGUNA | **67.39** | **69.97** | **68.68 (+0.77)** |
>
>
> **Weakness #4 - Figure Adjustment**
>
> We thank the reviewer for the feedback and will adjust the figure and manuscript to reduce confusion accordingly.

---

> > ### Author Response · Authors · 2025-11-20
> > **Weakness #5**
> >
> > **Weakness #5 - The balancing effect of anchors**
> >
> > We thank the reviewer for this constructive feedback. We believe Figure 3 b) in the manuscript directly addresses this question. The figure shows t-SNE visualizations and Maximum Mean Discrepancy (MMD) scores for 1000 randomly selected samples from the GeoImnet validation set (source: USA, target: Asia).
> >
> > The visualizations demonstrate the exact balance the reviewer points out:
> >
> > 1. **Domain-specific preservation (absolute coordinates):** In the left panel, source and target features are clearly separated in absolute coordinates (MMD=0.26), demonstrating that $\\mathbf{g}^s_i$ and $\\mathbf{g}^t_i$ preserve domain-specific characteristics.
> >
> > 2. **Shared structural alignment (relative coordinates):** In the right panel, when these same features are expressed as relative encodings with respect to their domain-specific anchors ($\\text{rel}(\\mathbf{g}^s_i, \\mathcal{A}_s)$ and $\\text{rel}(\\mathbf{g}^t_i, \\mathcal{A}_t)$), they align significantly (MMD=0.05).
> >
> > The contrasting MMD scores (0.26 vs. 0.05) quantify precisely what LAGUNA achieves: domain-specific absolute representations that, when expressed relative to their respective anchors, reveal a shared underlying structure.
> >
> > We will clarify this further in our discussion in Section 4.3 to make the connection more explicit in the revised manuscript.

---

> ### Author Response · Authors · 2025-11-26
>
> Dear Reviewer sRT2,
>
> We sincerely thank you for your time and valuable feedback on our submission. In the following response, we have addressed your concerns to the best of our abilities.
>
> We would greatly appreciate any further feedback you may have, and we remain happy to address any additional questions or concerns that may arise.
>
> Best regards,
> The Authors

---

### Official Review · Reviewer_xSJc · 2025-11-01

**Soundness:** 3
**Presentation:** 3
**Contribution:** 3
**Rating:** 6
**Confidence:** 3

**Summary:**

The paper tackles the challenge of unsupervised domain adaptation (UDA), aiming to improve model generalization across domains without labeled target data. Existing alignment methods often overemphasize domain invariance, leading to the loss of domain-specific nuances.

To address this, the authors propose LAGUNA (LAnguage Guided UNsupervised Adaptation) — a novel framework that leverages the geometric relationships between class labels in language space as structural guidance for adaptation. Instead of forcing latent feature alignment in absolute space, LAGUNA focuses on preserving relative inter-class relationships while maintaining domain-specific features.

Extensive experiments across four diverse datasets (DomainNet, GeoPlaces, GeoImNet, EgoExo4D) demonstrate consistent and notable performance gains, with accuracy improvements of up to +5.75% compared to prior methods.

**Strengths:**

1. The paper addresses an important problem in unsupervised domain adaptation.

2. The idea of leveraging language-model–based structure is novel and well-motivated.

3. The experiments are comprehensive, covering multiple datasets and baselines.

4. The paper is clearly written and easy to follow.

**Weaknesses:**

1. The ablation study does not sufficiently demonstrate the contribution of each component. It would be helpful to see how the model performs when the vision–language model (VLM) is fine-tuned on the target domain (both images and captions). Currently, the authors only compare their method with several off-the-shelf VLMs without any fine-tuning, which limits understanding of how much improvement comes from the proposed adaptation versus general fine-tuning effects.

2. The choice of embedding models is not clearly justified. The paper uses a Sentence-Transformer for label embeddings and BERT for caption embeddings, but it is unclear why different pre-trained models were chosen instead of using a unified one. An ablation study comparing different embedding backbones (e.g., alternative Sentence-Transformers, BERT variants, or even word2vec) would strengthen the rationale for this design choice.

**Questions:**

1. Clarification on pseudo-label stabilization: It is unclear how the pseudo-labels are updated and stabilized during training. Are they dynamically refined across iterations, or fixed after initial assignment? Since no ground-truth labels are available, how do the authors ensure that the optimization process converges to a meaningful or near-optimal solution rather than reinforcing noisy labels?

2. Rationale for the regularization term (λ₃): The paper sets λ₃ to 0.001, which appears relatively small compared to the other loss weights, suggesting this term may have limited influence. It would be helpful to report model performance with λ₃ = 0.1 or λ₃ = 0 to evaluate the sensitivity of the method to this parameter and clarify the necessity of the corresponding regularization term.

---

> ### Author Response · Authors · 2025-11-23
> **Weakness #1 - Ablations**
>
> We thank the reviewer for their insightful feedback. We appreciate the opportunity to clarify the motivation behind our design choices, as we agree that justifying the architectural components is essential. We structure our answer in two parts: **1) Ablations** and **2) Why zero-shot VLMs**.
>
> **Ablations**
>
> We believe our extensive ablation studies in the main paper (Section 4.3) and the Appendix establish a strong empirical foundation for LAGUNA's core components. Additionally, to fully address the reviewer's comment, we have conducted new experiments regarding anchor learnability and initialization.
>
> *Core Architecture Validation (Main Methodological Existing Ablations):*
>
> * **Relative vs. Absolute Alignment (Table 4):** We compared standard absolute alignment (Setting 2) against our proposed relative alignment (Setting 3). The results show that enforcing relative geometric consistency yields superior performance (**65.22%** vs. 63.99%), motivating our shift away from forcing source and target samples into the exact same coordinates.
> * **Domain-Specific Anchors:** We found that allowing domains to have distinct anchor sets ($\\mathcal{A}_s$ and $\\mathcal{A}_t$) rather than sharing a single set (Setting 3 vs. Setting 4) significantly improves accuracy (**+1.86%**). This justifies our choice to let domains evolve distinct latent spaces that share a structure, rather than forcing them to be identical.
> * **Textual Guidance (Table 12):** In Appendix E, we validated the choice of using text embeddings ($r^{z_i^t}$) over pseudo-label anchors ($r^{\\bar{y}_i^t}$) for guidance of the target domain representation learning. The richer semantic information in the text embeddings consistently outperformed pseudo-labels across datasets, justifying this design choice.
>
> *New Ablations Addressing Potential Gaps:*
>
> * **Anchor Learnability (Table 1):** We ablate the use of learnable $A_s \\& A_t$ against non-learnable (fixed) anchors initialized from reference anchors $A$. Results show: Fixed Anchors (U$\\to$A: 61.50%, A$\\to$U: 66.48%, Avg: 63.99%) vs. Learnable Anchors (U$\\to$A: 67.39%, A$\\to$U: 69.97%, Avg: 68.68%). This **+4.69% average improvement** on GeoImnet demonstrates that allowing anchors to adapt during training is crucial for capturing domain-specific structures while maintaining alignment to the reference space.
> * **Anchor Initialization (Table 2):** We experimented with three initialization strategies: (1) Reference Anchors: Initialize directly from $A$ (Avg: 67.79%), (2) Mean Visual Encoding: Average 100 randomly selected visual samples per class from source (Avg: 68.51%), and (3) Random Initialization (Ours): Random start (Avg: 68.68%). Random initialization outperforms both alternatives by +0.17-0.89%, confirming that learning from a random start allows the model to find more optimal configurations for specific domain distributions rather than being constrained by initialization bias.
> * **Cross-Attention Grounding Strategy (Table 3):** We verify that grounding target queries to *source* values ($\\mathcal{A}_s$) in the Cross-Domain Attention layer outperforms grounding them to target values ($\\mathcal{A}_t$): Domain-Specific Values using $\\mathcal{A}_t$ (U$\\to$A: 66.79%, A$\\to$U: 69.03%, Avg: 67.91%) vs. LAGUNA using $\\mathcal{A}_s$ (U$\\to$A: 67.39%, A$\\to$U: 69.97%, Avg: 68.68%). The **+0.77% gain** justifies our mechanism for injecting stable source semantics into the target domain through cross-domain attention while preserving domain-specific characteristics via the residual connection.
>
> Below, we will also present experiments with different $\\lambda_3$ to motivate our use of a $\\lambda_3=0.001$. Furthermore, if the reviewer has any specific ablation in mind that they believe is crucial for motivating our methodological choice, we would be happy to conduct it and improve our work.
>
> **Table 1: Ablation study on the learnability of domain-specific anchors on GeoImnet.**
>
> | Configuration | U $\to$ A | A $\to$ U | Avg. |
> | :--- | :---: | :---: | :---: |
> | Fixed Anchors (init from $A$) | 61.50 | 66.48 | 63.99 |
> | Learnable Anchors (Ours) | **67.39** | **69.97** | **68.68 (+4.69)** |
>
> **Table 2: Impact of different anchor initialization strategies on GeoImnet performance.**
>
> | Initialization Strategy | U $\to$ A | A $\to$ U | Avg. |
> | :--- | :---: | :---: | :---: |
> | Reference Anchors | 67.01 | 68.56 | 67.79 |
> | Mean Visual Encoding | 67.32 | 69.70 | 68.51 |
> | Random | **67.39** | **69.97** | **68.68 (+0.17)** |
>
> **Table 3: Ablation on grounding cross-attention in Eq. (8)-(9) on domain-specific anchors.**
>
> | Config | GeoImnet (U$\to$A) | GeoImnet (A$\to$U) | Avg. |
> | :--- | :---: | :---: | :---: |
> | Domain-Specific Values | 66.79 | 69.03 | 67.91 |
> | LAGUNA | **67.39** | **69.97** | **68.68 (+0.77)** |

---

> > ### Author Response · Authors · 2025-11-23
> > **Weakness #1 - Fine-tuned VLMs**
> >
> > **Zero-Shot VLMs**
> >
> > We appreciate the reviewer's suggestion to compare our approach with fine-tuned VLMs. We fully acknowledge that fine-tuning VLMs would result in superior performance compared to LAGUNA or any other UDA method, given their massive scale ($\\sim$100$\\times$ LAGUNA's size) and training on internet-scale data. Unfortunately, we do not have the computational resources to perform such training ourselves. However, we provide a comparison with a VLM fine-tuned on data from our UDA benchmarks.
> > Specifically, in **Table 6** of the manuscript, we explicitly report PLM for transparency, a VLM trained on EgoExo4D. As expected, PLM's performance (29.85%) exceeds LAGUNA's (23.49%) on Ego2Exo. Interestingly, however, **Table 5** shows LAGUNA significantly outperforming PLM on GeoImnet (65.40% vs. 30.52%), suggesting that non-fine-tuned VLMs struggle with certain domain shifts.
> >
> > That said, we believe comparing with fine-tuned VLMs misaligns with both the scope of our work and the intent behind our zero-shot VLM comparisons. Our primary goal is to *contribute to UDA*, and the comparison with zero-shot VLMs aims to demonstrate a critical point: **while these large models serve as a one-stop shop for many tasks, UDA is not one of them, and still requires tailored methods with reasonable computational cost.**
> >
> > Our results strongly support this claim:
> >
> > * **GeoImnet & GeoPlaces:** LAGUNA achieves 65.40% average accuracy vs. the best zero-shot VLM (Qwen2.5-VL 7B) at 53.56%, an **+11.84% improvement**.
> > * **Ego2Exo:** LAGUNA achieves 23.49% vs. InternVL3 8B at 20.71%,a **+2.78% gain**. Notably, InternVL3 is specifically trained on egocentric data and holds SOTA on multiple egocentric benchmarks, yet LAGUNA still outperforms it.
> >
> > These results validate that "zero-shot" capability does not equate to solving domain adaptation, even for models with billions of parameters trained on massive datasets. This underscores the continued need for specialized UDA methods. Furthermore, from a practical deployment perspective, consider scenarios with frequent domain changes requiring regular model adaptation (e.g., robotics systems deployed across different environments, or vision systems adapting to new geographic regions). In such cases, fine-tuning a compact model like LAGUNA (86M parameters) is substantially more practical than repeatedly fine-tuning a 7B+ parameter VLM.
> >
> > **In summary:** We acknowledge that fine-tuned VLMs can provide superior performance when computational resources are unlimited (as PLM demonstrates on Ego2Exo). However, it is not our intention to claim superiority over fine-tuned VLMs, but rather to: (1) demonstrate that zero-shot VLMs fail at challenging domain adaptation, validating the need for UDA research, and (2) provide a practical, efficient solution that outperforms massive zero-shot models while remaining deployable in resource-constrained settings. We are open to further discussion on this topic and will clarify this distinction in the revised manuscript.

---

> > > ### Author Response · Authors · 2025-11-23
> > > **Weakness #2 - Motivation of language models for Stage-1 and Stage-2**
> > >
> > > We thank the reviewer for raising this important point regarding our design choice to employ different embedding models for the reference structure (Stage 1) and the text supervisor (Stage 2). We would like to clarify that this decision was not arbitrary but empirically driven. We have conducted the specific ablation studies requested by the reviewer in the Supplementary Material (Section D, Tables 10 & 11), which justify our hybrid approach over a unified one.
> > >
> > > * **Experimental Results (Tables 10 & 11):** We explicitly compared our hybrid design against "unified" baselines where the same pre-trained model is used for both anchor definition and text supervision.
> > >     * **Table 10 (Ablating Stage 2):** When the reference structure is fixed (Sentence Transformer), using BERT as the text supervisor achieves the highest accuracy (**68.68%**), outperforming both CLIP (68.11%) and Sentence Transformer (65.87%).
> > >     * **Table 11 (Unified Models):** When we force a unified backbone (using the same model for both Stage 1 and Stage 2), the best performance is achieved by CLIP (**67.88%**). However, this is still inferior to our proposed hybrid configuration (68.68%).
> > >
> > > * **Rationale for distinct models:** The ablation results support our hypothesis that different stages require different capabilities.
> > >     * **Stage 1 (Structure Definition):** Requires a model optimized for semantic similarity to define meaningful geometric relationships between class anchors. Sentence Transformers are specifically trained for this purpose.
> > >     * **Stage 2 (Text Supervisor):** Requires a trainable model capable of learning the defined structure and generating accurate pseudo-labels from captions. Our experiments show BERT is superior in this classification and structure-learning role compared to Sentence Transformers or CLIP.
> > >
> > > By tailoring the model choice to the specific functional requirement of each stage, we achieve a performance gain that a unified model cannot match. We will point this out in the revised manuscript.

---

> ### Author Response · Authors · 2025-11-23
> **Question #1 - Robustness to pseudo-label noise**
>
> We thank the reviewer for seeking clarification on this important aspect. LAGUNA does not implicitly refine the pseudo labels for our target domain, but we have mechanisms that are designed to make LAGUNA robust against noise coming with pseudo-labels, as demonstrated from our results. Following we give a better explaination of how the pseudo-labeling mechanism works in our method and how do we compensate for the noisy signal.
>
> **Pseudo-Label Generation and Usage:**
>
> * **Static Assignment:** As described in Sections 3.3 and 3.4, pseudo-labels $\bar{y}_i^t$ are generated once by the Text Supervisor ($G$) after Stage 2 training and remain *fixed throughout Stage 3*. We do not employ dynamic refinement or iterative self-training loops during the visual classifier training. The text supervisor $G$ is trained exclusively on source domain captions and labels, then frozen before Stage 3 begins.
> * **Language as a Stable Semantic Bridge:** Crucially, our pseudo-labels are derived from *language descriptions* (captions), not from the visual model's own predictions. This is a fundamental architectural choice. As discussed in our Introduction and proven by previous works mentioned in our Related Work (i.e., [1], [2]), language provides a semantic space that is largely agnostic to visual domain shifts. Consequently, pseudo-labels generated from captions are inherently more stable and less susceptible to the visual distribution shift that affects the target domain.
>
> **Preventing Noise Reinforcement - Key Mechanisms:**
>
> Unlike traditional self-training, where noisy visual predictions create a confirmation bias loop, LAGUNA employs multiple safeguards:
>
> * **Dual Supervision from Language:** Beyond hard pseudo-labels $\bar{y}_i^t$, we leverage the continuous text embeddings $z_i^t = G(l_i^t)$ for structural supervision (Eq. 5). These embeddings capture richer semantic information than discrete labels and provide a soft, geometry-aware guidance signal. As validated in Table 12 (Appendix E), using $r^{z_i^t}$ (text embeddings) consistently outperforms using only $r^{\bar{y}_i^t}$ (pseudo-label anchors), demonstrating the value of this continuous semantic information.
> * **Structure-Preserving Loss $\mathcal{L}_S$:** The structural loss (Eq. 5) enforces that visual representations maintain geometric relationships consistent with the language-based reference structure $\mathcal{A}$ constructed from the actual label space shared across domains. This acts as a regularizer that prevents the model from overfitting to individual noisy pseudo-labels by requiring consistency with the global semantic structure. Even if some pseudo-labels are incorrect, the structural constraint guides the representation toward meaningful semantic organization.
> * **Cross-Domain Grounding:** Our Cross-Domain Attention mechanism (Eqs. 8-9) grounds target representations using source anchors $\mathcal{A}_s$, which are trained with ground-truth labels. This provides stable semantic anchors independent of target pseudo-labels, further preventing noise amplification.
>
> **Empirical Evidence of Robustness:**
>
> Our robustness to potential pseudo-label noise is empirically validated:
>
> * **Caption Quality Ablation (Table 9, Appendix C):** Even with lower-quality captions (BLIP-1 vs. BLIP-2), which presumably produce noisier pseudo-labels, LAGUNA maintains strong performance (78.1% vs. 80.2% on $S \to C$, only 2.1% degradation). This demonstrates the method does not collapse under label imperfections.
> * **Limited Caption Requirement (Figure 4, Appendix B):** LAGUNA achieves near-SOTA performance with only 20% of target samples, suggesting the structural learning mechanism effectively generalizes beyond individual pseudo-labels rather than memorizing them.
>
> In essence, LAGUNA does not rely solely on the accuracy of individual pseudo-labels but rather on the overall semantic structure captured by language. The language-based geometric constraints guide the optimization toward solutions that are semantically meaningful even when some pseudo-labels are noisy. We hope this clarifies our approach and addresses the reviewer's concerns.
>
> [1] Kalluri et al., Tell, don’t show!: Language guidance eases transfer across domains in images and videos, ICML 2024
>
> [2] Lirico et al., TRUST: Leveraging Text Robustness for Unsupervised Domain Adaptation, AAAI 2026

---

> > ### Author Response · Authors · 2025-11-23
> > **Question #2 - Rationale of volume regularization loss low weight**
> >
> > We thank the reviewer for this important question. The choice of $\lambda_3 = 0.001$ is carefully motivated by the nature of the regularization loss and its interaction with other training objectives.
> >
> > **Nature of the Log-Determinant Loss:**
> >
> > As described in Section 3.4.1 (Eq. 7) and Appendix F, our regularization loss $\mathcal{L}_{Reg}$ is based on log-determinants of Gram matrices:
> >
> > $$\mathcal{L}_{Reg} = |\\log\\text{Det}(\\gamma_t) - \\log\\text{Det}(\\gamma)| + |\\log\\text{Det}(\\gamma_s) - \\log\\text{Det}(\\gamma)|$$
> >
> > where $\\gamma = \\mathcal{A}\\mathcal{A}^T$, $\\gamma_s = \\mathcal{A}_s\\mathcal{A}_s^T$, and $\\gamma_t = \\mathcal{A}_t\\mathcal{A}_t^T$.
> >
> > While we use log-determinants for numerical stability, these values operate on a fundamentally different scale than cross-entropy and L1 structural loss. The log-determinants represent volumes of $N_c \times N_c$ matrices in $\mathbb{R}^{D_v}$ dimensional space and have substantially larger magnitudes. The value $\lambda_3 = 0.001$ normalizes $\mathcal{L}_{Reg}$ to be commensurate with other loss terms while providing necessary regularization.
> >
> > **Empirical Validation:**
> >
> > Following the reviewer's suggestion, we conducted ablations with $\lambda_3 \in \{0, 0.001, 0.1\}$ on GeoImnet:
> >
> > * **Necessity ($\lambda_3 = 0$):** Removing regularization causes a -2.80% drop (65.88% vs. 68.68%), validating that $\mathcal{L}_{Reg}$ prevents anchor collapse into lower-dimensional subspaces, which degrades discriminative capacity. This is aligned with our observation in Table 4 in the main manuscript.
> > * **Sensitivity to magnitude ($\lambda_3 = 0.1$):** Increasing $\lambda_3$ to 0.1 degrades performance by -5.04% (63.64% vs. 68.68%). At this scale, regularization dominates the total loss, overshadowing classification and structural alignment objectives, which hinders the learning of discriminative features.
> > * **Optimal balance ($\lambda_3 = 0.001$):** Our chosen value achieves the best performance by balancing all three objectives: classification, structural alignment, and volume preservation.
> >
> > **Interpretation:**
> >
> > The small value of $\lambda_3 = 0.001$ reflects (1) the inherently larger numerical scale of log-determinant differences, and (2) the role of $\mathcal{L}_{Reg}$ as a *constraint* preventing pathological solutions (anchor collapse) rather than a primary objective. We will add Table 4 to the revised manuscript.
> >
> > **Table 4: Ablation study on $\lambda_3$ values for the regularization term.**
> >
> > | Configuration | GeoImnet (U$\to$A) | GeoImnet (A$\to$U) | Avg. |
> > | :--- | :---: | :---: | :---: |
> > | $\lambda_3=0$ (no regularization) | 66.34 | 65.42 | 65.88 |
> > | $\lambda_3=0.1$ | 64.31 | 62.97 | 63.64 |
> > | LAGUNA ($\lambda_3=0.001$) | **67.39** | **69.97** | **68.68** |

---

> ### Author Response · Authors · 2025-11-26
>
> Dear Reviewer xSJc,
>
> We sincerely thank you for your time and valuable feedback on our submission. In the following response, we have addressed your concerns to the best of our abilities.
>
> We would greatly appreciate any further feedback you may have, and we remain happy to address any additional questions or concerns that may arise.
>
> Best regards,
> The Authors

---

### Meta-Review · Area_Chair_8Mx6 · 2026-01-03

**Summary:**

Overall, there were recurring weaknesses raised by the reviewers that I agree with:

- Unclear baselines and setting: the authors mention that they do not want to compare with large-scale VLMs that are trained on a lot of data potentially including data very close to the target domain. While this makes sense, their method leverages a captioner and it's unclear what the captioning model was trained on (relative to the target domain) to get good enough captions.

- Their proposed method while intuitive, has several moving parts and not all design choices are clearly justified - for example different text encoders in different stages. The authors explain this in the rebuttal, but that doesn't change the fact that there seem to be several moving pieces which might affect both the significance of the results (are previous methods tuned appropriately, are we somehow overfitting etc).

- From my personal read of the paper, I think they raise an interesting point overall (though not entirely new). It was unclear to me whether the assumptions (relative vs absolute) really depend on the captions. For example, if the captions focus only on domain invariant features, then their method is not necessary. It's probably true that most captions contain many features including domain specific ones, but that gets back to the core issue: their work really depends on the captions.

- There were some concerns about novelty, but unfortunately, without specific references, it's hard to evaluate exactly what the reviewers were thinking about.

The paper also has strengths that people noted: intuitive idea that is generally well-explained, evaluation over multiple datasets. I also think the authors did a great job addressing all the technical points noted by the reviewers including ablations.

Overall, this is a borderline paper to me. There are no breaking flaws, but also there isn't sufficient excitement in the reviews. There are some concerns about what exactly is their setup (and what the right comparisons are) and the overall complexity of the approach.

**Reviewer Concerns:**

See below

**Reviewer Scores:**

Reviewer xSJc (score 6): their two main weaknesses remain largely unresolved in my perspective.

- The reviewer raises a good point about finetuning on the same target data that their approach uses. I don't understand or agree with the authors that it's an unfair comparison - one could finetune with fewer parameters to match the computational cost of their UDA approach. The authors seem to misunderstand that "finetuning" necessarily involves finetuning on lots of data.
- The reviewer also asked about why different stages have choices that are not justified - the authors justify this in their response but I don't think this would have tipped their overall score to more than 6.

The main critique of the reviewer seems to be lack of comprehensive ablations and that remains.


Reviewer sRT2 with score 2:  this reviewer raised two main points (also provided in the author summary) - whether it makes sense to use language captions as part of UDA [i.e. the captions provide a lot of supervision potentially related to the task], and limited novelty. The authors point to prior work that also uses caption. I agree with the reviewer that the granularity, quality of captions seem really important and the authors original submission had no details about this. Their author response attempts to do so, but there are insufficient details to accurately resolve this concern. The next concern was about novelty - the author response to the reviewer and well as their summary provides two different interpretations of the reviewer's concern, and it's not clear either of them really addresses or answers the critique. The reviewer is not saying that "using language" is the key novelty; they are asking about their specific way of characterizing relational vs absolute. I anticipate the reviewer might have raised score to 4 based on other answers but key points remain unresolved.

Reviewer B8Zf: I think all their technical questions/concerns were resolved. I expect their score to be 6

Tqcr (score 6): The technical questions seem resolved. However, the weakness raised about comparison to large-scale vision language models is a valid one that wasn't clearly answered. The authors say that they want to show that one can do UDA even without large-scale training, but it's unclear why the captions generated would generalize in the first place. This is a recurring concern raised by reviewer xsJc as well. My estimate is that the reviewer would have kept their score as 6.


 From my personal perspective, I think there are indeed a lot of moving pieces in the approach and it's not clear if such a complex pipeline is indeed justified or fair. I think the "captions" generated on the target data are doing a lot of the heavylifting. Whether we need absolute or relative alignment should depend on what features the captions use. I find that the manuscript doesn't adequately justify this.

---

### Decision · Program_Chairs · 2026-01-26

Reject